# Word learning from a tablet app: Toddlers perform better in a passive context

**Lena Ackermann**[1,2☯¤]*, **Chang Huan Lo**[3☯], **Nivedita Mani**[1,2], **Julien Mayor**[4]

**1** Psychology of Language Department, University of Göttingen, Göttingen, Germany, **2** Leibniz Science Campus "Primate Cognition", Göttingen, Germany, **3** School of Psychology, University of Nottingham Malaysia, Semenyih, Malaysia, **4** Department of Psychology, University of Oslo, Oslo, Norway

☯ These authors contributed equally to this work.
¤ Current address: Donders Institute for Brain, Cognition and Behaviour, Radboud University, Nijmegen, The Netherlands
* lenaackermann@gmx.net

**Data Availability Statement:** The data that support the findings of this study are publicly available at OSF, https://osf.io/rntjc/?view_only=2c0f72479380421fa3e15a2b363cb62c.

## Abstract

In recent years, the popularity of tablets has skyrocketed and there has been an explosive growth in apps designed for children. Howhever, many of these apps are released without tests for their effectiveness. This is worrying given that the factors influencing children's learning from touchscreen devices need to be examined in detail. In particular, it has been suggested that children learn less from passive video viewing relative to equivalent live inter-action, which would have implications for learning from such digital tools. However, this so-called video deficit may be reduced by allowing children greater influence over their learning environment. Across two touchscreen-based experiments, we examined whether 2- to 4-year-olds benefit from actively choosing what to learn more about in a digital word learning task. We designed a tablet study in which "active" participants were allowed to choose which objects they were taught the label of, while yoked "passive" participants were presented with the objects chosen by their active peers. We then examined recognition of the learned associations across different tasks. In Experiment 1, children in the passive condition outperformed those in the active condition (n = 130). While Experiment 2 replicated these findings in a new group of Malay-speaking children (n = 32), there were no differences in children's learning or recognition of the novel word-object associations using a more implicit looking time measure. These results suggest that there may be performance costs associated with active tasks designed as in the current study, and at the very least, there may not always be systematic benefits associated with active learning in touchscreen-based word learning tasks. The current studies add to the evidence that educational apps need to be evaluated before release: While children might benefit from interactive apps under certain conditions, task design and requirements need to consider factors that may detract from successful performance.

**Funding:** The authors received no specific funding for this work.

**Competing interests:** The authors have declared that no competing interests exist.

## Introduction

Within a few years of the iPad's debut, the popularity of touchscreen devices has skyrocketed. British and American households with children have seen more than a ten-fold increase in tablet ownership in the last years (British: 7% (2010) to 89% (2019); American: 8% (2011) to 78% (2017); [1–3]), with at least 42% of American children reported to own their own tablet in 2017 [3]. In parallel with this surge in tablet popularity, there has been an explosive growth in apps. To date, the Apple App Store features nearly 200,000 apps for education [4] and many of these are targeted at children, with toddlers or preschoolers being the most popular age category [5]. Most of these apps are released without prior formal evaluation [6] and only few apps aimed at preschoolers provide developmentally appropriate guidance and feedback [7]. Yet, at least 80% of parents of 2- to 4-year-olds report having downloaded apps for their children [3].

What remains in doubt is whether children learn from such apps on touchscreen devices or from electronic screen media given that young children exhibit reduced learning from passive video viewing and benefit more from equivalent live experience. This reduced learning, referred to as the "video deficit effect" [8], has been demonstrated in various tasks, including word learning [9, 10], in which children have been passively exposed to training stimuli on a screen, e.g., where they were given no choice in what they were being trained on and by whom.

The video deficit effect can be mitigated by providing children with a more interactive learning context. For instance, the provision of socially contingent feedback on infants' and toddlers' behavior has been shown to improve performance in object retrieval [11]; action imitation [12] and word learning tasks [13, 14]. Similarly, Lauricella, Pempek, Barr, and Calvert [15] suggest that this deficit may also be mitigated with pseudo-social contingent computer interactions, i.e., where children interacted with the game and could steer the course of the actions presented in the game by providing user input (e.g., pressing particular buttons).

Furthermore, the kind of pseudo-social contingency employed differentially impacts children's performance in a tablet-based object retrieval task. In particular, specific-contingency (requiring children to tap on a specific location on the screen) supported learning in younger 2-year-olds but not in older children, who instead benefited more from passive video watching [16]. Kirkorian, Choi, and Pempek [17] suggest that specific-contingency provides younger toddlers (23.5 to 27.5-month-olds) with the required attentional support to encode target features in complex scenes [18–20], while the same contingent experience disrupts learning in older toddlers (32 to 36-month-olds). For instance, Russo-Johnson, Troseth, Duncan, and Mesghina [21] found no main effects of different contingency situations (watch, tap or drag objects introduced to children within a word-learning app) among 2- to 4-year-old children, with all children learning words within the app. Taken together, the results on the effects of pseudo-social contingency on learning appear to be mixed across ages and the different types of contingency tested.

The above studies have focused on interactivity in a controlled context, in that the participants had no control over what they were to learn. A further way to involve participants in a more active learning situation is to allow participants to choose the *kind* of information to be learned. In adults, such active learning situations lead to superior performance relative to situations in which information is passively encountered [22, 23]. These findings have been extended and replicated in studies involving children [24–26]. For instance, Sim, Tanner, Alpert, and Xu [26] find that 7-year-olds who had control over their learning experience performed better than those who had new information presented in a random manner. Similarly, Ruggeri, Markant, Gureckis, and Xu [25] found that giving active control to 6- to 8-year-olds in a simple memory game enhanced their recognition memory and the advantage persisted in

the follow-up test held a week later, while Begus, Gliga, and Southgate [27] found that letting 16-month-olds decide what information to receive facilitated their performance in an imitation task.

Another study–more similar to the present study–examined the effects of selective learning in 3- to 5-year olds using a tablet-based word learning task [24]. Children in the active condition were given control over the *order* in which 15 toys were labelled, whereas those in the receptive condition could tap on a button in the center of the screen to hear the labels (in a pre-specified order). The testing phase, consisting of tests of children's recognition of 1, 2, 4, and 8 toys in separate blocks, revealed that selective learning improved information retention, and could be attributed to the increased level of engagement. However, since the improvement was observed only in the earlier blocks, which tested fewer word-object associations, it is difficult to tell whether the effect only occurred early in learning or whether the complexity of the blocks involving more objects overshadowed the reported effect. In addition, participants in this task could not select the kind of information they could learn, i.e., *which* of a selection of objects they would rather hear the label for. They could only determine the *order* in which objects were labelled.

Some recent studies have provided children with the choice of which objects they could choose to be given more information about and the influence of such choice on learning. For instance, Zettersten and Saffran [28] presented children with either fully ambiguous, partially ambiguous or disambiguated word-object mapping situations. Here, children could choose to hear the label of an object that would resolve the ambiguity in ambiguous mapping situations. In cases where the relative ambiguity of the pairs presented was increased, children did show some evidence of preferentially selecting the object that would resolve the ambiguity. This suggests that children actively choose objects that can reduce their information gap at least at the older ages tested in these studies (four to seven years of age, see also [26]).

In the present studies, an app was designed to teach younger children novel words in a yoked design, i.e., either via active selection (where children could decide *which* objects they could hear the label for) or passive reception (where selections were made for them, based on the choices made by yoked age-matched children in the active condition). To control for overall exposure during the learning phase, the study was designed such that the sequence, exposure time, and content of the learning phase were held constant across each yoked active-passive pair. We examined word learning in the context of two identification tasks, namely a two-alternative forced choice task and a four-alternative forced choice task. We tested a wide age range of children across ages (24-months, 30-months and 40-months) that have been targeted in previous studies suggesting differences in the influence of active learning on performance [17]. This allowed us to investigate the developmental time course of the impact of active learning on word learning. Based on this previous work on the effects of interactivity in learning, we expected improved performance in the active condition relative to the passive condition at the younger age groups and the opposite pattern in the older age groups, although we note that this prediction contrasts with findings of an active benefit in older children's word learning [24].

## Experiment 1

### Materials and methods

**Participants and design.** A total of 130 German-speaking children took part in the study, with 42 participants in the 24-month-old group and 44 participants in each of the 30- and 40-month-old groups. Mean age, age range and standard deviation for each age group are detailed in Table 1. Yoked age-matched pairs of children (ages at date of testing within 2

**Table 1. Mean age, standard deviation and age range for all three age groups.**

| Age group | $M_{age}$ (months) | $SD_{age}$ (months) | $Range_{age}$ (months) |
|---|---|---|---|
| 24 months | 24.31 | 1.16 | 22.05–25.96 |
| 30 months | 29.81 | 1.49 | 28.16–35.22 |
| 40 months | 39.69 | 3.52 | 36.01–47.97 |

months of each other) were assigned to either the active or the passive condition. In the active condition, participants could select four novel objects to be told the label of, while in the passive condition, participants were automatically given the labels for the objects chosen by their yoked active peers. An additional pair of participants in the 24 month-old group had to be excluded due to missing data and an additional two pairs of participants in the 30-month-old group had to be excluded due to a clear side preference in selection, i.e., tapping eight times consecutively on the image shown on a particular side, and inattentiveness, i.e., getting up and walking around during the study. The study was reviewed and approved by the ethics committee of the Georg Elias Müller Institute of Psychology, University of Göttingen. Caregivers gave written consent to their child's participation in the study.

**Apparatus and materials.** The study was carried out using an iPad Pro with a web application based on the framework provided in [29]. Images of 8 novel objects and 6 familiar objects were chosen for the experiment (see Figs 1 and 2) and child-directed speech was used in all audio recordings played. Vocabulary development norms suggest that over 75% of all 24-month-olds and close to 100% of all 30-month-olds already produce the six familiar words [30, 31] We chose four novel words as labels of the chosen objects: *Batscha*, *Foma*, *Kolat*, and *Widex*. These words follow the phonotactic constraints of German (see S1 Appendix for further details).

**Procedure.** The study began with a learning phase followed by a familiarisation phase and two test phases.

**Learning phase.** *Active condition.* The learning phase consisted of four trials and each trial began with a prompt asking the participant to select one of the two randomly combined images of the novel objects placed on the left and right sides of the screen respectively. In the first trial, the prompt was "Guck mal, hier sind zwei Bilder. Du kannst auf eines drücken." [Look, here are two pictures. You can tap on one.] For subsequent trials, the prompt was "Drück mal auf ein Ding, dann hörst du seinen Namen." [Tap on an object, then you'll hear its name.] Tapping was only enabled 300 ms after the prompt had ended to ensure that the tap could reliably be interpreted as a response to the presentation of stimuli. Upon tapping, a red outline was shown around the selected image while that which was not selected was hidden. The selected novel object was then labelled five times in the same trial using various carrier phrases, including: (a) "Guck mal, ein X!" [Look, a/an X!], (b) "Das ist ein X!" [This is a/an X!], (c) "Wow, da ist ein X!" [Wow, there is a/an X!], (d) "Siehst du das X?" [Do you see the X?], (e) "Toll! Das ist ein X!" [Great! This is a/an X!], where X was the novel word. All auditory stimuli were recorded by a female native speaker of German in child-directed speech. The time taken by the participant to make their selection was automatically recorded to be used to time stimulus presentation for the passive peer so that both participants saw the images for exactly the same amount of time. The subsequent trial began 1500 ms after the labelling had ended. In each trial, the pairs of novel objects displayed and the novel word given to this object were generated at random with no repeats. Thus, at the end of the learning phase, the participant was presented with four distinct novel labels for their chosen four novel objects.

*Passive condition.* Passive learning participants were not required to do anything but watch and listen as they would be exposed to the active learning peer's selections following the exact

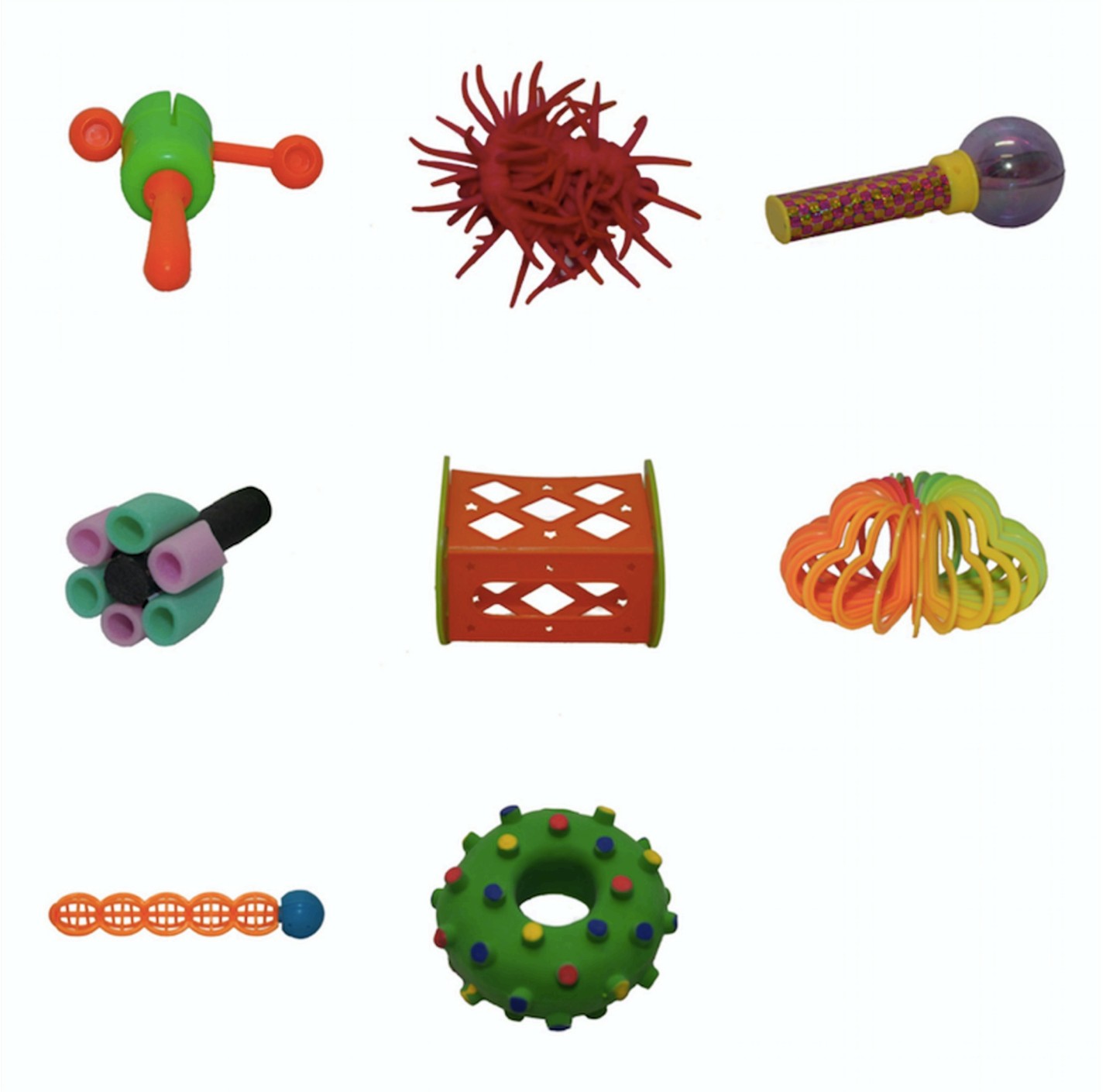

**Fig 1. Novel objects (taken from [41]).**

timing of the age-matched active peer. Here, tapping was disabled throughout the learning phase. Instead of being prompted to select something, an introductory audio: "Siehst du die zwei Bilder? Sind sie schön?" [Do you see the two pictures? Are they beautiful?] was played to attract the participant's attention to the images. The participant had to wait for as long as the active peer took to select between the two novel objects displayed before the selection was

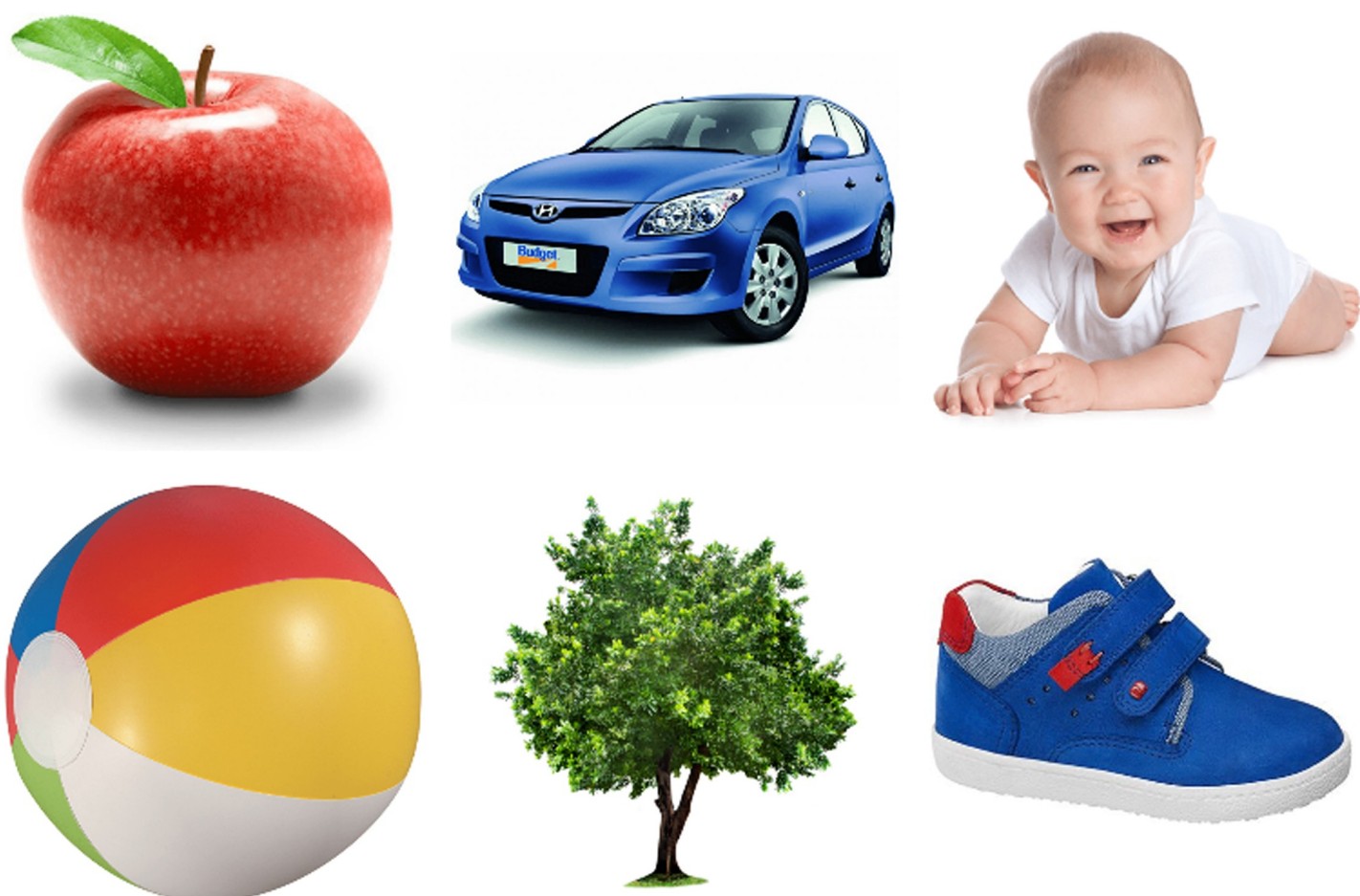

**Fig 2. Familiar objects.**

outlined in red and the unselected object was hidden. Participants in the passive condition heard the objects labelled in the same manner as the active condition, which repeated the novel word five times. A 1500 ms paused followed before the subsequent trial began. The order of the learning trials was identical to that which had been given to the active peer.

**Familiarisation phase (all participants).** Based on results of pilot tests with 42 pairs of children aged between 18 and 60 months, 6 familiar trials were included, following the learning phase, to: (a) familiarise the passive group with tapping, and (b) keep the participants engaged. Each familiar trial consisted of a pair of randomly generated familiar objects, also placed on the left and right sides of the screen respectively, followed by the label for one of these objects embedded in a carrier phrase, e.g., "Drück mal auf den Schuh" [Tap on the shoe]. No feedback was given after their response regardless of which object the child tapped on. The response and time taken to respond were recorded. There was a 1500 ms pause before the subsequent trial began.

**Two-alternative forced choice rest phase (all participants).** This phase consisted of 12 two-alternative forced choice trials where each novel word was tested (paired separately with each of the three other objects) three times, in counterbalanced order. In each trial, participants were shown two of the four novel objects which they had heard labels for previously and asked to tap on the object associated with the heard novel word. There was no time limit to

respond and each trial ended with a 1500 ms pause. As with the familiar trials, the response and time taken to respond were recorded.

**Four-alternative forced choice test phase (all participants).** This phase consisted of 8 four-alternative forced choice trials where each novel word was tested twice, in counterbalanced order. In each trial, participants were shown all four novel objects which they had learnt labels for and asked to, once more, tap on the object associated with the heard novel word. The images of the novel objects were positioned randomly in a 2x2 grid on the screen. There was also no time limit for the participants to respond and each trial ended with a 1500 ms pause. Again, the participants' responses and reaction times were recorded.

## Analysis and results

**Reaction time.** Reaction time was measured in ms from the onset of the target word. As there was no time limit for responses, responses included outliers as high as 1014 s (in one case where the participant got up, played with something else, then returned to make their selection and continue with the experiment). A Shapiro-Wilk normality test (W = 0.082, p < 0.01) suggested that the data did not follow a normal distribution. The data was therefore log transformed prior to further analysis. To make sure only those trials where the child was engaged in the task were included in our analysis, outliers were removed using a criterion of 2 SDs above the mean. The number of outliers decreased with increasing age (72 at 24-months of age (35 active, 37 passive), 42 at 30-months of age (20 active, 22 passive) and 31 at 40-months of age (14 active, 17 passive)) with roughly equal number of outliers in each condition. Unadjusted and adjusted mean reaction times and standard deviations for each age group are detailed in Table 2.

Fig 3 shows the distribution of children's reaction times in each phase, split by age and condition. We fitted linear mixed-effects models (LMMs) to assess whether reactions times differed across conditions (active vs. passive) in each of the three test phases. The results of the mixed-effects models are detailed in Table 3. The model included the interaction between condition (active vs. passive) and age as fixed effects, and selected object and participant pair as random effects. Condition (*passive* = 1, *active* = -1) and age were sum-coded, resulting in the following model:

$$RT_{log} \sim Condition*Age + (1|Participant\ pair) + (1|Object)$$

In addition to the intercept-only model reported above, we also ran models with a maximal random effects structure (see S3 Appendix). The main results obtained across both kinds of models were similar with regard to the accuracy analyses reported below but not with regard to the response time results reported here (deviations are flagged in every instance they occurred). We chose to focus on the model reported here for the following reason: we report

**Table 2. Mean reaction times and standard deviations before (unadjusted) and after (adjusted) outlier removal, split by age group.**

| Age group and condition | Unadjusted $M_{RT}$ (s) | Unadjusted $SD_{RT}$ (s) | Adjusted $M_{RT}$ (s) | Adjusted $SD_{RT}$ (s) |
|---|---|---|---|---|
| 24 months active | 4.86 | 7.52 | 3.35 | 2.74 |
| **24 months passive** | 6.96 | 44.07 | 3.43 | 2.83 |
| 30 months active | 4.57 | 10.30 | 3.26 | 2.37 |
| **30 months passive** | 4.40 | 7.04 | 3.35 | 2.30 |
| 40 months active | 3.26 | 2.85 | 2.99 | 2.05 |
| **40 months passive** | 3.30 | 3.87 | 2.74 | 1.82 |

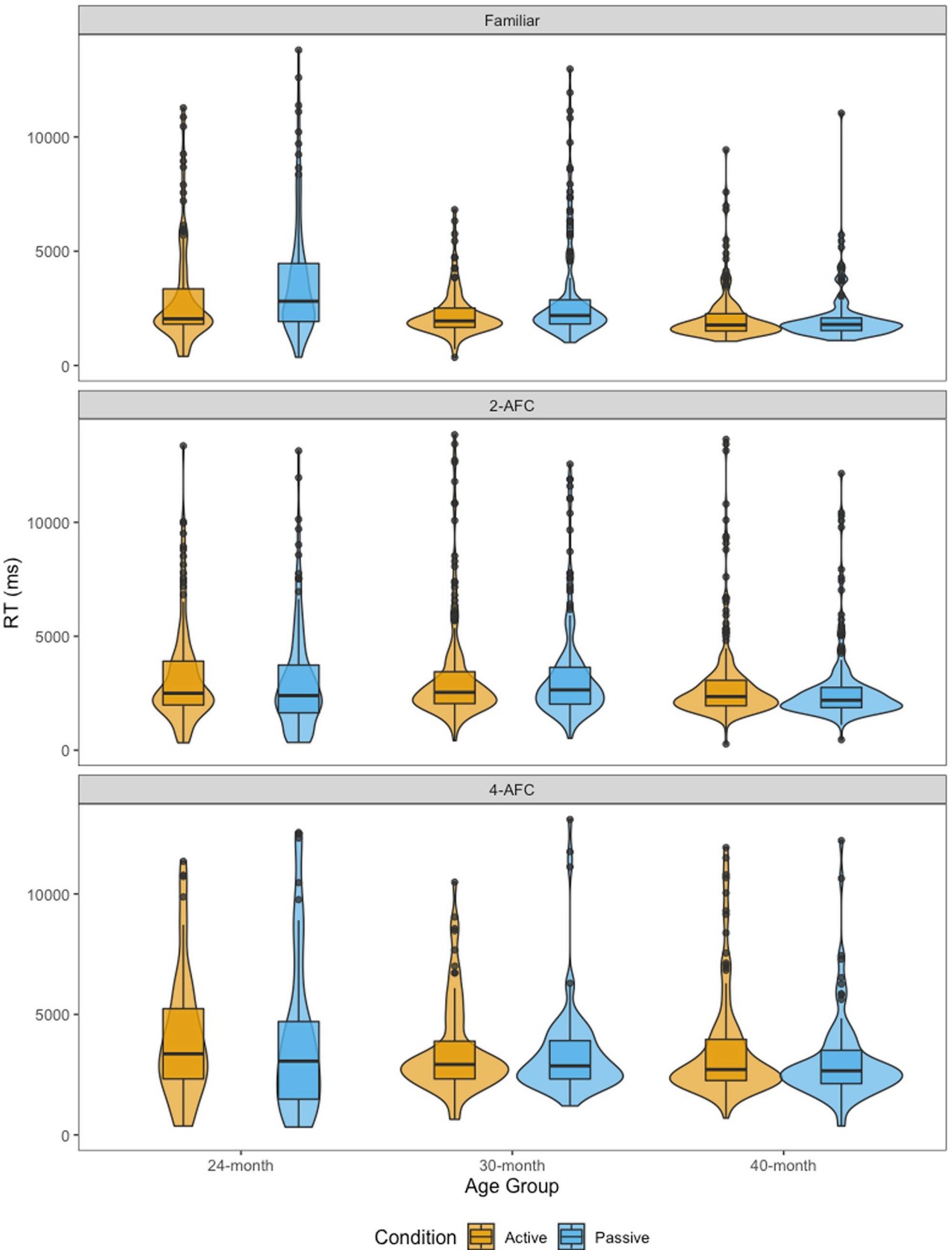

**Fig 3. Reaction time by phase.** Reaction time for correct responses in the familiar, 2-AFC and 4-AFC test phases, split by age group and condition, with outliers (> 2 SD) removed.

results from a 2-AFC and a 4-task, where the object is a target in one trial is a distractor in the other trial. Performance on a given trial is therefore confounded with performance in a different trial. We therefore opted against a complex random effect structure to reduce this confound. Nevertheless, for transparency, we also report the models with this more complex random effect structure in S3 Appendix.

As Table 3 suggests, children in the active condition were faster to tap the target object relative to children in the passive condition in familiar object trials and the effect of condition interacted with age. This is potentially due to the latter being required to tap on the screen for the first time in these trials. Indeed, B1 Fig in S2 Appendix suggests, it was only in the first few trials that there was a difference between children in the active and passive condition for familiar trials and this difference was reduced later on in the experiment. In contrast, in the 2-AFC trials, we found that children in the passive condition were faster than children in the active condition overall, and that the effect of condition interacted with age (Note that this was not the case in the maximal model reported in S3 Appendix. We will therefore treat this result with some caution). There was no effect of condition in the 4-AFC task, nor were there any interactions between condition and age in this task. In other words, there was a difference in the effect of condition between 24- and 40-months in the familiar model and between 24- and 30-months in the 2-AFC model, but not in the 4-AFC model. Breaking the data by age, we found that there was a significant effect of condition at 24-months ($\beta$ = 0.126, p = .006) and 30-months ($\beta$ = 0.113, p < .001) in the familiar trials, with children from both age groups responding faster when they were assigned to the active condition. No such effect of condition was found in the 40-month age group ($\beta$ = -0.014, p = .537). With regard to the 2AFC trials, a significant effect of condition at 24-months ($\beta$ = -.091, p = .032) was found, with 24-month-olds responding faster when they were assigned to the passive condition relative to the active condition (Again, this was not the case in the maximal model). No such effect of condition was found in the older age groups (30-m: $\beta$ = 0.009, p = .747, 40-m: $\beta$ = -.035, p = .072).

**Accuracy.**    Fig 4 shows children's mean accuracy in identifying the labelled object in each phase. Binomial generalized linear mixed-effects models (GLMMs) with a logit-link function were used to analyze children's accuracy in the three phases. The models included the interaction between condition (active vs. passive) and age as fixed effects and selected object and participant pair as random effects. Condition (*passive* = 1, *active* = -1) and age were sum-coded,

**Table 3. LMM results for reaction time with condition $^*$ age interaction split by test phase.**

|  | Familiar | | | 2-AFC | | | 4-AFC | | |
|---|---|---|---|---|---|---|---|---|---|
|  | **β** | **SE** | **p** | **β** | **SE** | **p** | **β** | **SE** | **p** |
| (Intercept) | **7.734** | **0.024** | **< .001** | **7.849** | **0.026** | **< .001** | **7.986** | **0.056** | **< .001** |
| Condition | **0.076** | **0.019** | **< .001** | **-0.039** | **0.016** | **.019** | -0.044 | 0.025 | .076 |
| Age - 30m | 0.014 | 0.034 | .671 | **0.081** | **0.037** | **.028** | 0.043 | 0.070 | .532 |
| Age - 40m | **-0.168** | **0.033** | **< .001** | -0.022 | 0.036 | .537 | -0.010 | 0.068 | .888 |
| Condition$^*$ Age - 30m | 0.037 | 0.026 | .160 | **0.048** | **0.023** | **.036** | 0.034 | 0.034 | .322 |
| Condition$^*$ Age - 40m | **-0.089** | **0.026** | **< .001** | 0.004 | 0.022 | .868 | -0.033 | 0.033 | .312 |

Bold script indicates significance at p < 0.05.

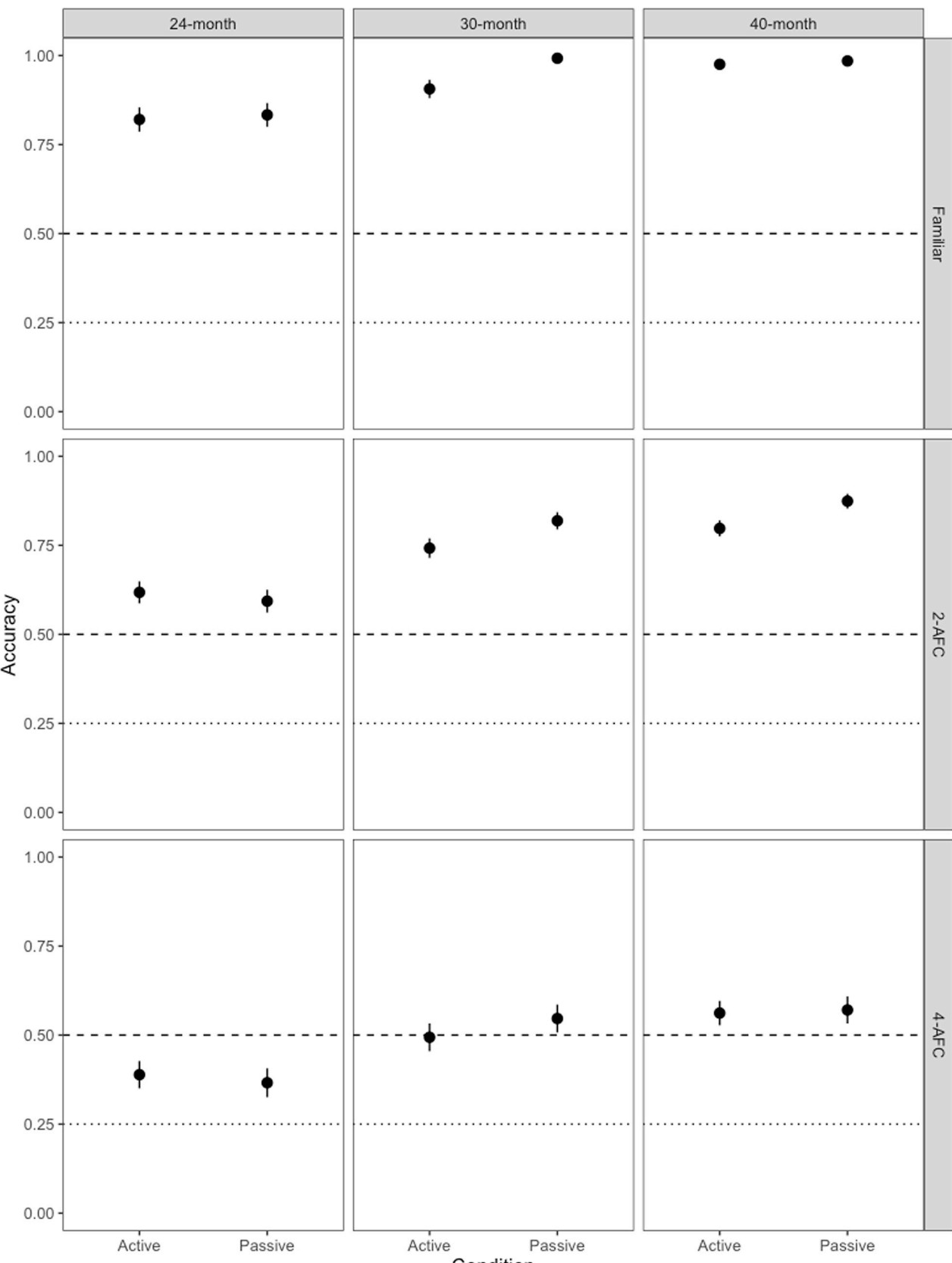

**Fig 4. Accuracy by phase.** Accuracy in the familiar, 2-AFC and 4-AFC test phases, split by condition and age. Dashed line represents chance (.5) in the familiar and 2-AFC test phases, dotted line represents chance (.25) in the 4-AFC test phase.

resulting in the following model:

$$\text{Accuracy} \sim \text{Condition} * \text{Age} + (1|\text{Participant pair}) + (1|\text{Object})$$

The results of the models are detailed in Table 4 (for results with a maximal random effects structure, see S3 Appendix).

In the familiar phase and the 2-AFC, condition significantly predicts accuracy, with children providing more accurate responses in the passive condition relative to the active condition. There was a difference in the effect of condition between 24- and 30-months in the familiar model. The effect of condition in the familiar phase and the interaction between condition and age (at 24- and 30-months) was not significant in the maximal model (S3 Appendix). There were no interactions between condition and age between any of the other ages tested in any of the other models. There was no effect of condition on accuracy in the 4-AFC task. Note that the intercept was not significant in the 4-AFC model given that chance here cannot be set artificially to .25 (as ought to be the case in a 4-AFC task). Breaking down the interactions between condition and age, we found a significant difference between children in the active and children in the passive condition for familiar trials only at 30-months of age ($\beta$ = 1.296, p = .014) but not at 24-months ($\beta$ = 0.024, p = .890) or at 40-months ($\beta$ = 0.267, p = .558). Despite the fact that we did not find a significant interaction between condition and age in the 2-AFC model, we broke down the data by age to examine potential developmental differences. Noting that this analyses must be treated with caution (given the non-significant interaction), we found a significant difference in the accuracy of children's response in the active and the passive condition at 30-months ($\beta$ = 0.242, p = .031) and at 40-months ($\beta$ = 0.279, p = .020) but not at 24-months ($\beta$ = -.057, p = .552).

## Discussion–experiment 1

In this experiment, we set out to examine whether being given the opportunity to choose the objects that will be labeled influences children's learning of these word-object associations in a touchscreen-based word learning task. Children were assigned to either an active learning task, where they were allowed to choose the objects they would hear the label of or a yoked passive learning task, where they would hear the label of an object an age-matched yoked active child had chosen.

**Table 4. Model results for accuracy.**

|  | Familiar | | | 2-AFC | | | 4-AFC | | |
|---|---|---|---|---|---|---|---|---|---|
|  | **β** | **SE** | **p** | **β** | **SE** | **p** | **β** | **SE** | **p** |
| Intercept | **3.184** | **0.278** | **< .001** | **1.183** | **0.093** | **< .001** | -0.033 | 0.114 | .773 |
| Condition | **0.527** | **0.236** | **.025** | **0.153** | **0.063** | **.016** | 0.018 | 0.067 | .785 |
| Age - 30m | 0.564 | 0.404 | .162 | 0.189 | 0.131 | .147 | 0.139 | 0.122 | .252 |
| Age - 40m | **0.949** | **0.370** | **.010** | **0.565** | **0.133** | **< .001** | **0.339** | **0.120** | **.005** |
| Condition* Age - 30m | **0.784** | **0.384** | **.041** | 0.086 | 0.090 | .343 | 0.091 | 0.094 | .333 |
| Condition* Age - 40m | -0.280 | 0.346 | .418 | 0.126 | 0.094 | .178 | -0.016 | 0.091 | .856 |

Bold script indicates significance at p < 0.05.

First, we discuss the responses during the familiar test phase where children were asked to tap on one of two familiar objects whose label they were presented with. Here, we found that active children responded faster relative to children in the passive condition. This finding suggests that the active children likely responded faster because they had more experience tapping on objects while the familiar test phase was the first point in the experiment where passive children were asked to tap on the screen. This is especially likely since the difference between the active and passive children appears to be constrained to the first two trials in the task with overlapping reaction times from the third trial onwards at least at 24- and 30-months of age (see B1 Fig in S2 Appendix). Thus, while there appears to be an active benefit in the recognition of familiar objects, this appears to be artefact of the task and the experience that children in the two groups had with tapping objects on the screen. With regard to the accuracy of children's responses, we found a passive boost, with older children (30-month-olds) responding more accurately in passive condition relative to the active condition, but no such passive boost at the younger or the older age-group. Even with the 30-month-olds, this appears to be limited to the first trial and not to later trials. Especially with regard to the older age groups, we note that responding is at ceiling (see Fig 4). Given this pattern of responding, we suggest that differences between active and passive children in the familiar test phase are treated with caution.

Next, we discuss the results of the 2-AFC task, i.e., where children were asked to tap on one of two novel objects whose label they had been presented with. Here we found differences across conditions across the three ages tested, although these differences varied across the two measures and the different models reported. In particular, we found that 24-month-old children who were assigned to the passive condition were faster to recognize the target relative to children who were assigned to the active condition. No differences in reaction times were found at the older age groups and in the maximal models reported. With regard to the accuracy measure, we found a main effect of condition which did not interact with age, suggesting no developmental differences in the passive benefit across the ages tested. Nevertheless, exploratory analyses found that 30- and 40-month-olds who were assigned to the active condition responded with decreased accuracy relative to the yoked passive children at the same ages. No such difference in accuracy was found in the younger age-group. Neither did we find differences across conditions in the 4-AFC task. In other words, collapsing across the measures, we find no developmental differences in the passive benefit across the ages tested here (but see [17, 21]). With regard to the different measures, we find a reaction time boost in the younger children and an accuracy boost in the older children, with both of these measures favouring the passive children relative to the active children.

Although this result is congruent with other studies showing improvement in performance for children assigned to a passive condition relative to conditions including pseudo-social contingency [17], the findings raise a number of questions. In particular, what remains uncertain is whether the differences found here relate to differences in the performance of children across the two conditions relative to differences in competence. In other words, do children assigned to the active condition merely perform worse than their passive counterparts while nevertheless having learned the words to an equal degree or do children assigned to the active condition also learn worse than their passive counterparts? For instance, one explanation for the poorer performance of the active children may be that they continue to choose the objects that they like (as during the learning phase of the experiment) rather than choosing the objects whose label they have been presented with, despite having learned the novel word-object associations. Clarification of the competence-performance distinction is therefore required before further interpretation of the results is possible. Experiment 2 examined this issue in further detail using a more implicit measure of children's eye-movements as they completed the task. If active children learn worse than their passive counterparts, we would expect poorer

performance, i.e., less accurate fixations to the target object, even on such an implicit measure. On the other hand, if poorer performance of the active children is due to their not conforming to the demands of the task, we would expect similar performance across active and passive children as indexed by the looking time measure.

Experiment 2 therefore attempted to replicate the results of Experiment 1 while extending this using an additional implicit looking time measure (similar to the preferential looking tasks used in laboratory studies). In addition, we tested Malay-speaking children from Malaysia in Experiment 2 allowing us to examine the extent to which the findings replicate in children from a different cultural and linguistic background.

## Experiment 2

### Participants and design

Thirty-two typically developing, primarily monolingual Malay-speaking children, aged between 28 and 35 months ($M$ = 30.25, $SD$ = 1.71, range = 27.59–34.76) participated in the study. Yoked age-matched pairs of children (ages at date of testing within half a month of each other) were assigned to either the active or the passive condition. As in Experiment 1, in the active condition, the participants could select 4 novel objects to be told the labels of, while in the passive condition, the participants were automatically given the labels for the objects chosen by their active, age-matched peers. Due to a clear side preference in selection, i.e., tapping 8 times consecutively on the image shown on a particular side ($n$ = 3), and inattentiveness, i.e., getting up and walking around during the study ($n$ = 3), an additional six pairs of participants had to be excluded from analysis. The study was reviewed and approved by the Science and Engineering Research Ethics Committee (SEREC) of the University of Nottingham Malaysia. Caregivers gave written consent to their child's participation in the study and webcam video recording of their child during the study.

### Apparatus and materials

The study was carried out using a Microsoft Surface Pro 3 tablet with a web application that captures both a participant's implicit (gaze)—with the device's built-in webcam which was facing the participant—and explicit (tapping) responses. Images of eight novel objects and six familiar objects were chosen for the study (see Experiment 1). All auditory stimuli used were recorded by a female native speaker of Malay in child-directed speech. We selected four disyllabic, novel words to be used as labels for the chosen novel objects: *banung*, *ifi*, *mipo*, and *pafka*. These words obeyed the phonotactic constraints of Malay (see S1 Appendix for further details).

### Procedure

The design was identical to Experiment 1 with the only difference being the language in which the stimuli were presented and that webcam videos of the participants were recorded for the entire duration of the study.

**Learning phase.** *Active condition.* The learning phase was set up identically to that of Experiment 1, with the only difference being the language in which the prompts were produced. Thus, in the first trial, the prompt was "Tengok ni, sini ada dua gambar. Pilih satu." [Look, here are two pictures. Pick one.] For subsequent trials, the prompt was "Pilih satu gambar, lepas tu kita akan dengar nama dia." [Pick a picture and then we'll hear its name]. Upon tapping, the selected novel object was then labelled five times in the same trial using various carrier phrases, including: (a) "Tengok, X!" [Look, a/an X!], (b) "Ini adalah X!" [This is a/an

X!], (c) "Wow, itu X!" [Wow, that is a/an X!], (d) "Nampak tak X?" [Do you see the X?], and (e) "Bagus! Ini adalah X!" [Great! This is a/an X!], where X was the novel word.

*Passive condition.* Passive learning participants were not required to do anything but watch and listen as they would be exposed to the active learning peer's selections according to the exact timing of the age-matched active peer. Auditory prompts were in Malay with "Nampak tak dua gambar tu? Cantik kan?" [Do you see the two pictures? Beautiful, right?] presented in the first trial, and in subsequent trials "Mari kita dengar nama untuk gambar lagi." [Let's hear names for pictures again] to attract the participant's attention to the images. All other details were identical to Experiment 1.

**Familiarisation phase.** As in Experiment 1, six familiar trials were included. In each familiar trial, participants were presented with a pair of familiar objects, followed by the instruction to tap on one of these objects based on a given label X embedded in the carrier phrase "Tunjukkan gambar X." [Show (me) the picture of X.].

**Two-/four-alternative forced choice test phase (2-AFC/4-AFC).** All details of the design for the 2-AFC and 4-AFC task were identical to Experiment 1, with the exception being that the auditory prompts were in Malay (see carrier phrase from Familiarisation phase).

## Gaze analysis

In addition to participants' explicit responses, participants' viewing behaviour was also recorded in all trials, including trials in the learning phase. To quantify this, each video was split into 200 ms chunks, as had been done in [32] on the basis that saccades take approximately 200 ms to initiate [33]. These video chunks were presented in a random order to the two raters who were to rate them as: (a) "left", when the participant was looking to the left side of the screen; (b) "right", when the participant was looking to the right side of the screen; (c) "away", when the participant was looking away from the screen; or (d) "indeterminable", when none of three other options applies (see Fig 5 for examples). To avoid potential biases, rating was carried out in a blind rating situation where the position of the target was unknown to the rater.

We report, as is standard in the literature, the proportion of target looking to the images during both the training and the test phases. In addition, we plot the looking time during the learning phase and the test phase in Figs 6, 7 and 8. Together these capture not just the proportion of looks to the target but also the look-aways to the distractor since the proportion of target looks would correspondingly drop at any given time were the child to be looking at the distractor rather than the target. Ten percent of the video chunks were rated by one of the authors and a research assistant. Calculating Cohen's Kappa, we found a substantial agreement [34] between the two raters overall, $\kappa = 0.705$ (79.7% agreement). Upon excluding video chunks which were coded as "indeterminable", an almost perfect agreement was found, $\kappa = 0.950$ (97.1% agreement). When only differentiating between "left" and "right", agreement rose to 99.2%, $\kappa = 0.984$. Thus, it can be inferred that both raters agreed on the side of the screen participants were looking at, when they were able to decide on one.

Following coding, for further analysis, the target was set as the object that was labelled in both the learning trials and the test trials. Based on these ratings, the proportion of looks to the target in each trial was computed. As it was not feasible to rate 4-AFC trials, only the learning trials, the familiar trials and the 2-AFC trials were analysed here.

In the learning phase, participants' viewing behaviour was coded for all four learning trials from the onset of the labelling for the selected novel object, i.e., right after a selection was made. We used looking time during the learning phase as a predictor in the model examining learning in the 2-AFC and 4-AFC test phases to account for differences in attention to the

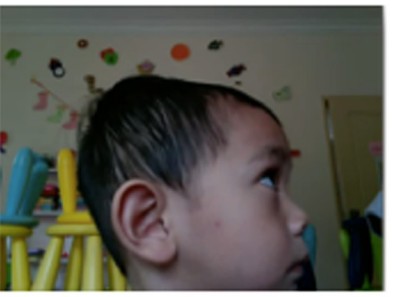

away

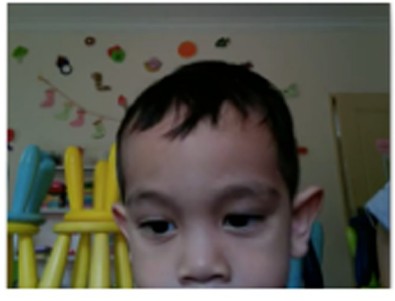

right

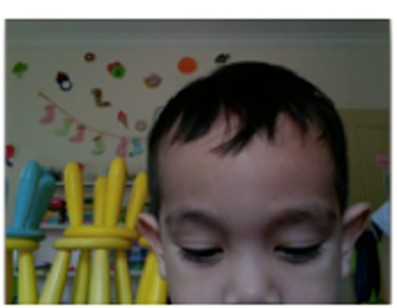

indeterminable

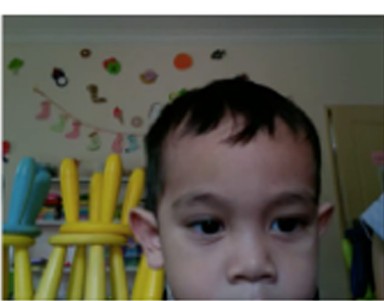

left

**Fig 5. Video rating scale.** Each video was split into 200 ms chunks and rated as either looking at the "left" or "right" (side of the screen), "indeterminable", or "away" (from the screen). Written consent for publication of the participating child's pictures was obtained from the caregiver.

labelled object during learning across conditions. In the familiar phase and in the 2-AFC trials, participants' viewing behaviour was coded from the onset of the presented target word to when participants chose the target object.

To examine potential differences between the active and the passive participants' gaze patterns over the course of the learning trials, the familiar trials, and the 2AFC trials, we conducted three cluster permutation analyses for each of these trial types [35–38]. The first compared the average proportion of looks at the target between the two conditions (active vs. passive), whereas the second and third compared the average proportion of looks at the target in each condition to the chance level (50%; active vs. chance and passive vs. chance). For the test trials analyses we only considered fixations that occurred between 200 to 2000 ms post target word onset in the familiar trials and 400 to 2200 ms post target word onset in the 2-AFC trials to minimize the effect of motor planning. We shifted the time window for the 2-AFC trials as children take longer in mapping newly learned words than familiar words [39, 40].

Prior to the analyses, we removed trials where more than 25% of the video chunks were rated as "indeterminable". This retained 113 of 128 trials from all 32 participants in the learning phase, 182 of 184 trials from all 32 participants in the familiarisation phase, and 311 trials from 31 participants of 372 trials from 32 participants in the 2-AFC test. All proportions of target looks were arcsine-root transformed to better fit the assumptions of the $t$-test conducted at each time point to compare the proportions of target looks to chance or between the two conditions. Time points with a significant effect ($t > 2$, $p < .05$) were then grouped into a cluster, for which its size was obtained from the summation of all $t$-values within this cluster. To test the significance of a cluster, we conducted 1000 simulations where conditions (active vs. passive, active vs. chance, passive vs. chance) were assigned randomly for each trial and obtained

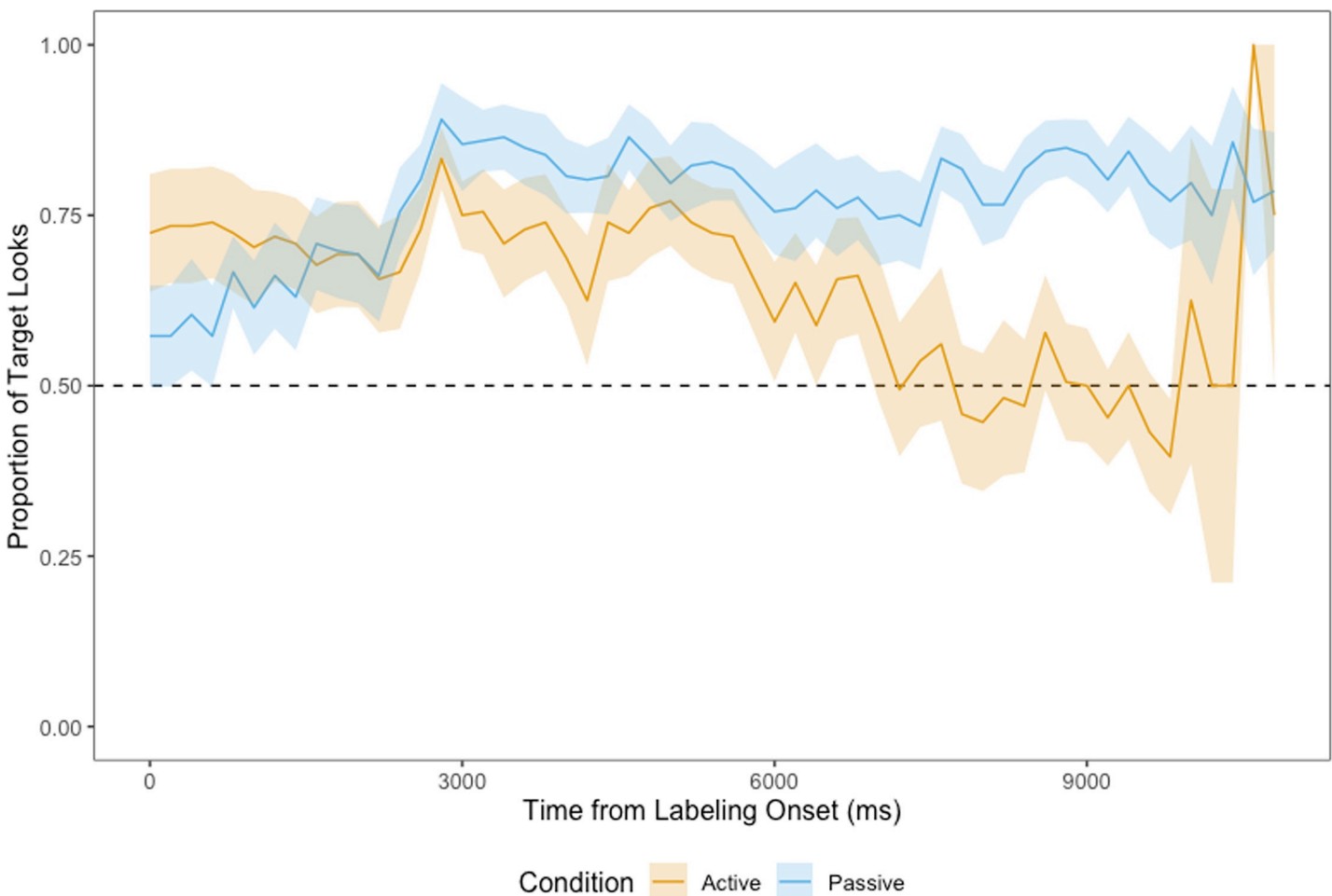

**Fig 6. Proportion of looks at the target in the learning phase, time-locked to the labelling of the selected novel object.** Dashed line represents chance (0.5).

the size of the biggest cluster in each simulation using the same procedure as before, with the real data. If the probability of observing a cluster—from the randomised data—with the same size as or bigger than the cluster from the real data was smaller than 5% ($p < .05$), the cluster from the real data was considered significant, i.e., the differences (active vs. passive, active vs. chance, passive vs. chance) were significant.

## Analysis and results

**Gaze data.** *Learning phase.* Fig 6 shows children's proportion of looks at the target in all 4 learning trials, from the onset of the labelling of the selected novel object. Children in the passive condition looked more at the target than children in the active condition overall and the cluster-based permutation analysis led to the identification of a significant difference across conditions from 7600 ms to 9800 ms following onset of the label ($p = .001$). Children in the passive condition fixated the target significantly above chance (0.5) for most of the duration of the 10 s labelling phase (from 1600 ms to 10000 ms, $p < .001$), while their active peers fixated the target significantly above chance (0.5) for the first-half of the labelling phase (from 0 ms to 2000 ms, $p = .007$; from 2600 ms to 4000 ms, $p = .006$; from 4400 ms to 5600 ms, $p = .018$).

*Familiarisation phase.* Fig 7 shows children's proportion of looks at the target from the onset of the target word in familiar trials. The cluster-based permutation analysis revealed no

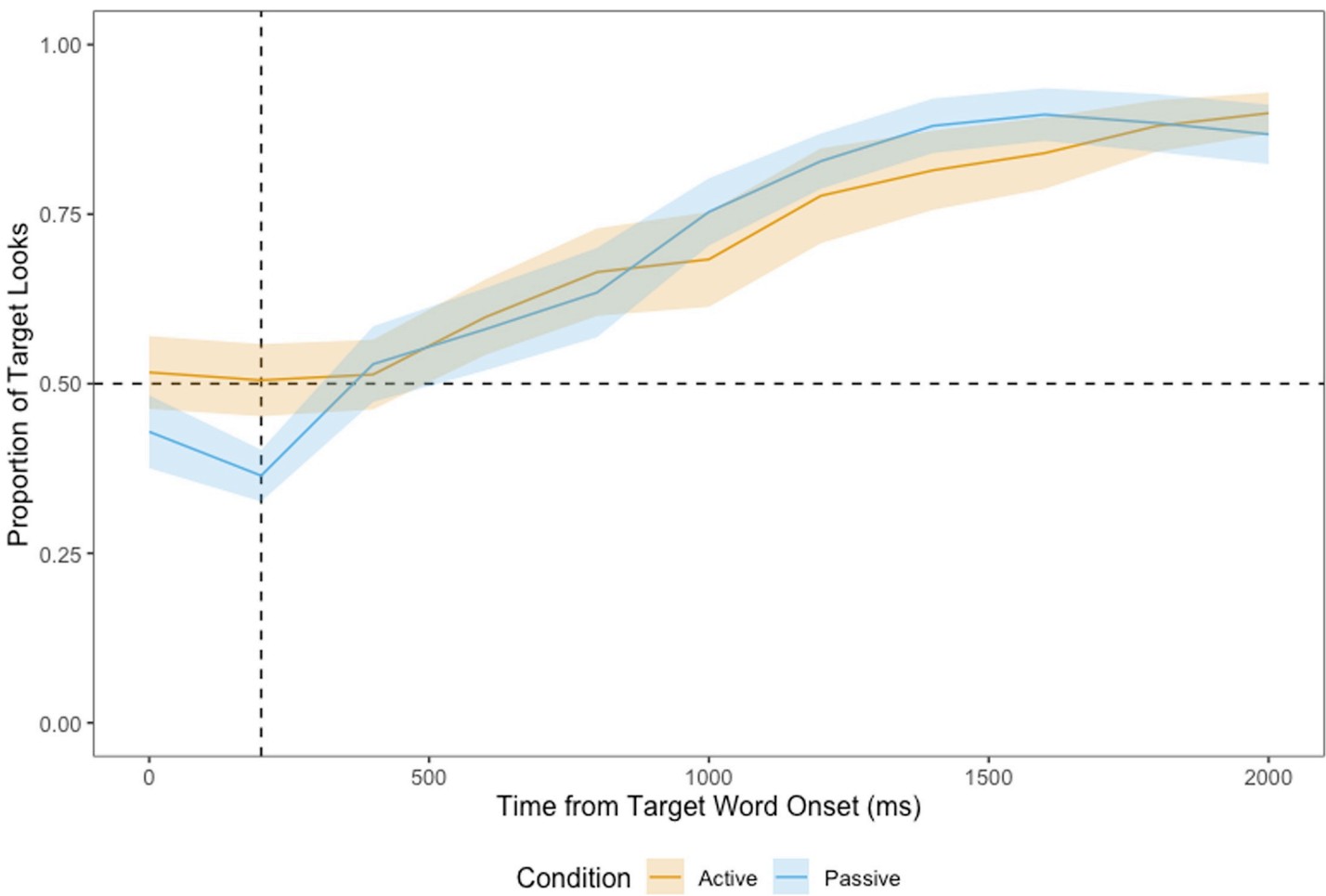

**Fig 7. Proportion of looks at the target in familiar trials, time-locked to the onset of the target word.** Dashed vertical line at 200 ms marks the beginning of the analysis window, dashed horizontal line represents chance (0.5).

timepoints where we found a significant difference between the active and the passive conditions. Children from both conditions fixated the target significantly above chance (0.5) shortly after the target word onset (from 800 ms to 2000 ms, $p < .001$).

*2-AFC test phase*. Fig 8 shows children's proportion of looks at the target from the onset of the target word in 2-AFC trials. The cluster-based permutation analysis revealed no timepoints where we found a significant difference between the active and the passive conditions. Children in the active condition fixated the target significantly above chance (0.5) shortly after the target word onset (from 800 ms to 2200 ms, $p < .001$). On the other hand, no significant timepoint was identified for children in the passive condition, although a one sample t-test across the entire time window indicated that they looked significantly above chance, $t(160) = 1.928$, $p = .028$.

**Reaction time.** Reaction time was measured in ms from the onset of the target word. As participants were not given a time limit to respond, responses included outliers as high as 114 s. Since the data did not follow a normal distribution as indicated by a Shapiro-Wilk normality test ($W = 0.495$, $p < .001$), the data was log transformed prior to further analysis. To ensure that only those trials where the child was engaged in the task were included in our analysis, we removed trials (28 active, 17 passive) using a criterion of 2 SDs above and below the mean.

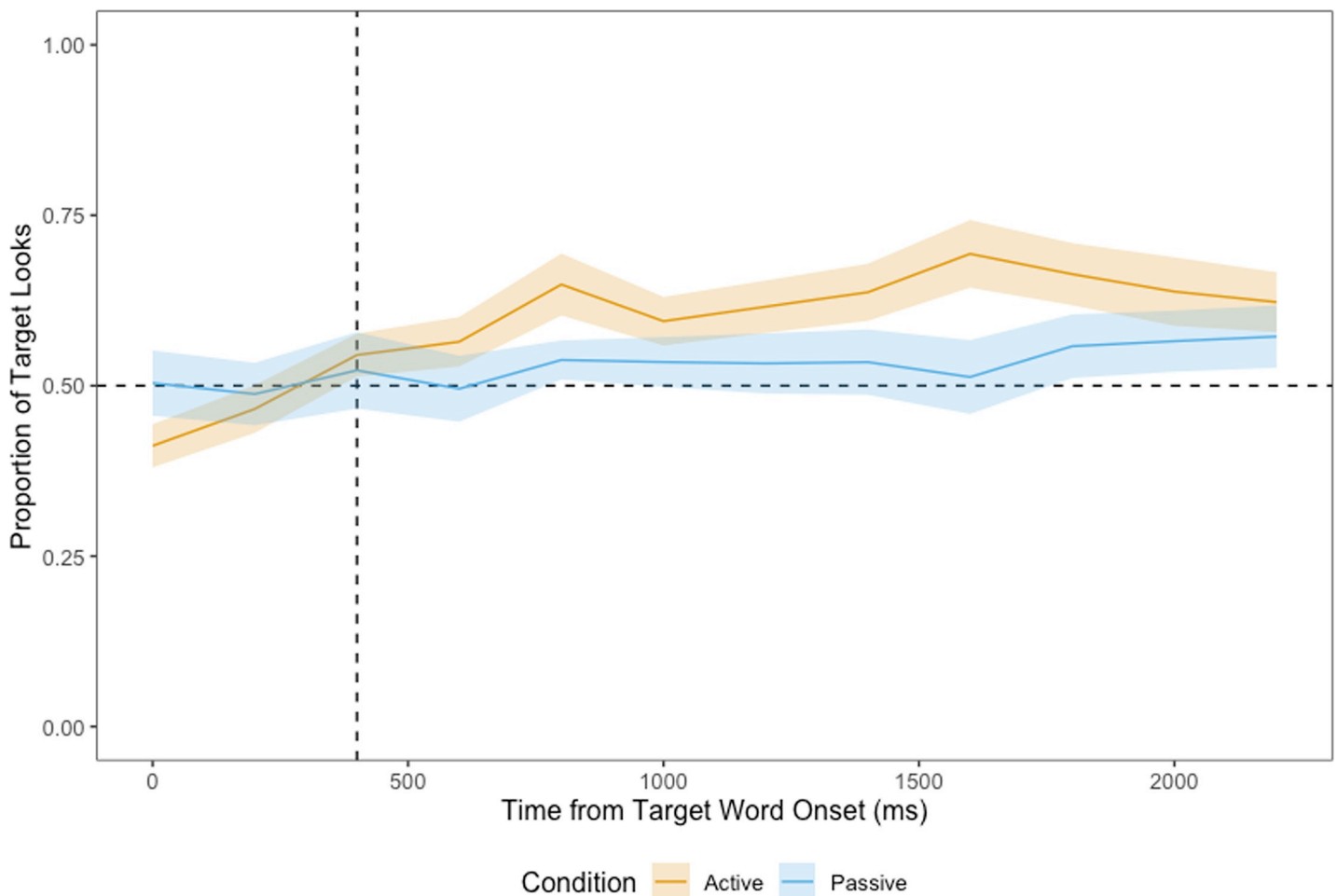

**Fig 8. Proportion of looks at the target in 2-AFC trials, time-locked to the onset of the target word.** Dashed vertical line at 400 ms marks the beginning of the analysis window, dashed horizontal line represents chance (0.5).

Unadjusted and adjusted mean reaction times and standard deviations for each condition are detailed in Table 5.

We fitted LMMs to assess whether reactions times differed across conditions (active vs. passive) in each of the three test phases. The results of the mixed-effects models are detailed in Table 5. The model included condition (active vs. passive) and learning looks (proportion of looks at the target during the learning phase; applicable to 2-AFC and 4-AFC only) as fixed effects and selected object and participant pair as random effects. Condition (*passive* = 1, *active* = -1) was sum-coded, resulting in the following model:

$$\text{RT}_{\log} \sim \text{Condition} + \text{Learning looks} + (1|\text{Participant pair}) + (1|\text{Object})$$

As in Experiment 1, in addition to the intercept-only model reported above, we also ran the model with a maximal random effects structure (see S3 Appendix).

**Table 5. Mean reaction times and standard deviations before (unadjusted) and after (adjusted) outlier removal, split by condition.**

| Condition | Unadjusted $M_{RT}$ (s) | Unadjusted $SD_{RT}$ (s) | Adjusted $M_{RT}$ (s) | Adjusted $SD_{RT}$ (s) |
|---|---|---|---|---|
| Active | 4.67 | 7.06 | 4.11 | 3.45 |
| Passive | 5.34 | 6.48 | 4.61 | 3.85 |

**Table 6. LMM results for RT.**

|  | Familiar | | | 2-AFC | | | 4-AFC | | |
|---|---|---|---|---|---|---|---|---|---|
|  | β | SE | p | β | SE | p | β | SE | p |
| (Intercept) | **8.080** | **0.123** | **< .001** | **8.100** | **0.164** | **< .001** | **9.002** | **0.331** | **<0.001** |
| Condition | 0.069 | 0.048 | .149 | 0.035 | 0.043 | .411 | 0.143 | 0.081 | .078 |
| Learning looks | - | - | - | 0.039 | 0.203 | .848 | **-1.232** | **0.428** | **.004** |

Bold script indicates significance at p < 0.05.

As Table 6 suggests, children in both the active and passive conditions did not differ in terms of speed in responding. While children in the passive condition were slower in the first few trials of each test phase, they eventually caught up with children in the active condition (see Fig 9). A significant effect of learning looks was found in the 4AFC test phase, with children who spent more time fixating the target during the learning phase being quicker in responding.

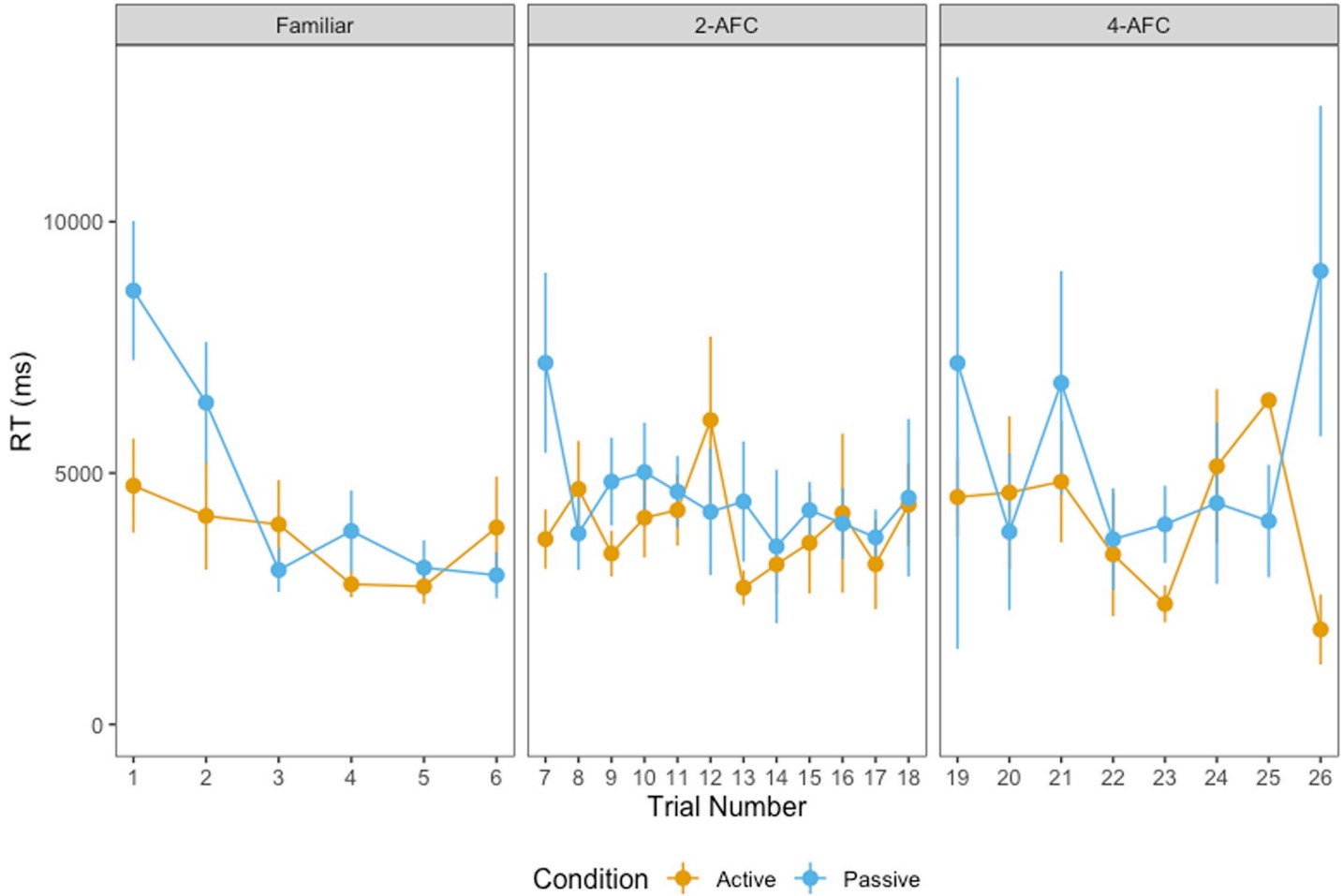

**Fig 9. Reaction time by trial.** Reaction time for correct responses in the familiar, 2-AFC and 4-AFC test phases, split by condition, with outliers (< 2SD and > 2 SD) removed.

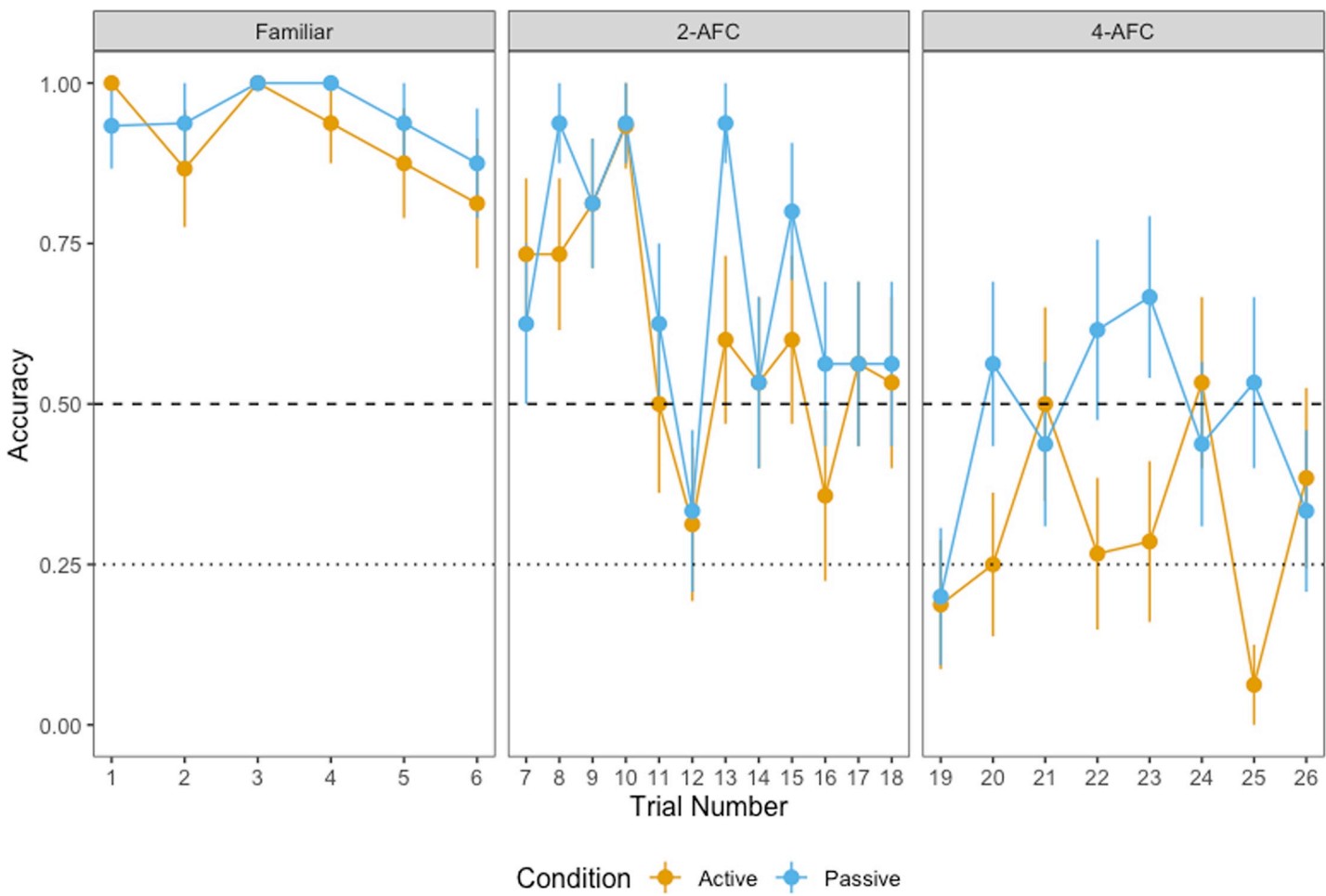

**Fig 10. Accuracy by trial.** Accuracy in the familiar, 2-AFC and 4-AFC test phases, split by condition. Dashed line represents chance (.5) in the familiar and 2-AFC test phases, dotted line represents chance (.25) in the 4-AFC test phase.

## Accuracy

Fig 10 shows children's accuracy in identifying the labelled object in each condition across the three test phases.

Binomial GLMMs with a logit-link function were used to analyze children's accuracy in the three phases. The models included the interaction between condition (active vs. passive) and learning looks (proportion of looks at the target during the learning phase; applicable to 2-AFC and 4-AFC only) as fixed effects and chosen object and participant pair as random effects. Condition (*passive* = 1, *active* = -1) was sum-coded, resulting in the following model:

$$\text{Accuracy} \sim \text{Condition} + \text{Learning looks} + (1|\text{Participant pair}) + (1|\text{Object})$$

The results of the models are detailed in Table 7.

There was no effect of condition on accuracy in the familiar phase. However, in both the 2-AFC and 4-AFC test phases, condition significantly predicted accuracy, with children providing more accurate responses in the passive condition relative to the active condition. Proportion of looks at the target during the learning phase was not a significant predictor in both critical test phases.

**Table 7. Model results for accuracy.**

| | Familiar | | | 2-AFC | | | 4-AFC | | |
|---|---|---|---|---|---|---|---|---|---|
| | β | SE | p | β | SE | p | β | SE | p |
| Intercept | **2.627** | **0.295** | **< .001** | **0.989** | **0.393** | **.012** | **-1.220** | **0.500** | **.015** |
| Condition | 0.264 | 0.295 | .371 | **0.246** | **0.115** | **.032** | **0.290** | **0.142** | **.041** |
| Learning looks | - | - | - | -0.549 | 0.531 | .302 | 1.035 | 0.666 | .120 |

Bold script indicates significance at p < 0.05.

## Discussion–experiment 2

Experiment 2 set out to replicate the findings of Experiment 1 with children from a different cultural background while also examining a more implicit measure of recognition performance–namely looking times data–across the two conditions. We found a very similar pattern of responding to the results with same-aged children from Germany and Malaysia (30-months). In particular, we found no difference between children assigned to the passive and active condition with regard to reaction times, but did find that children in the passive condition responded with greater accuracy than children in the active condition, in the 2-AFC task. The Malaysian children also demonstrated a passive advantage in the 4-AFC task. Note that the maximal model (provided in S3 Appendix) provided qualitatively similar results–in having a passive advantage in both the 2AFC and 4AFC conditions. There were no differences with regard to performance in the familiar trials across the two conditions.

Interestingly, the analysis of children's gaze behavior in the learning phase revealed that participants in the passive condition fixated on the labelled target object significantly longer and more robustly than their active counterparts, suggesting that children in the passive group may engage more with the learning material. One possible explanation for this pattern is that the design of the learning phase set the stage for different learning experiences across conditions: Active children, who are allowed to tap from the very beginning, have a more game-like experience than their passive counterparts, who are only allowed to tap later. Passive children might thus take the task more seriously, resulting in taking more time to encode the word-object-association. Alternatively, it may also be that active children have already explored the objects in depth before making the choice and once their choice is made need no longer examine this object in further detail, while passive children may reengage with the target object once this object has been presented as the target.

Nevertheless, analysis of children's performance in the tapping task revealed that gaze duration during the learning has no significant effect on children's accuracy in the test phase: While children in the passive condition looked longer at the target object in the learning phase *and* outperformed their active counterparts in terms of accuracy, the former did not predict the latter. Neither did we find differences in children's gaze behavior across the conditions in the 2-AFC task. This is particularly revealing given that we did find differences in children's accuracy in the 2-AFC and 4-AFC tasks. Taken together, we found no evidence that passive children's increased engagement with the learning material could explain their improved performance in the accuracy task and no evidence for a difference in children's eye movement behavior with regard to their accuracy and speed of identification of the target object. In other words, using our implicit measure, we find no difference between children assigned to the passive condition and the active condition. We discuss the implications of these results in more detail below.

## General discussion

In recent years, tablet ownership in families with children has increased drastically [3] and parents can choose from a large number of educational apps that claim to boost children's learning. However, as a majority of these apps have not been evaluated before release [6], many may fall worryingly short of their pledge.

The current studies aimed to bring together recent debates on active learning and learning from interactive media. They set out to explore how active selection of learning experiences affects word learning from a touchscreen-based app in toddlers aged 22 to 48 months. Children were assigned to either an active or a yoked passive condition. In the active condition, children were allowed to choose the object they wanted to hear the label of and then tested on their recognition of the novel word-object associations using both a touchscreen task (Experiments 1 and 2) and implicit gaze data (Experiment 2). In Experiment 1, we found differences in reaction times to tap the labelled object across children assigned to the passive and the active conditions (although we note that this was not significant in the maximal model), as well as differences in children's accuracy of identification of the target object across conditions. Across all ages tested here, we found a passive boost with children in the passive condition showing faster responses to target (24-months) or greater accuracy in target recognition (across all ages in the main model and separately at 30- and 40-months in exploratory analyses).

This apparent passive advantage may either be explained by a competence or a performance deficit with regard to the active children. The competence deficit explanation would suggest that interacting with the app by tapping may take up valuable cognitive resources. Children in the passive condition–who do not have to allocate resources to tapping–have more capacity to encode and retain the information presented to them. Here, the active children may actually learn and encode the novel word-object associations worse than the passive children. The performance deficit explanation would suggest that children in the passive condition may approach the task differently relative to children in the active condition. Participants in the active condition are allowed to tap their preferred object during the learning phase. Thus, they might treat the test phase as an extension of the learning phase and continue to merely indicate their preference for one of the objects during the test phase as well. Relatedly, tapping might be a prepotent response for children in the active condition, such that, instead of paying attention to the prompt, they might be waiting for their next chance to tap and do so as soon as they can, regardless of instruction. This interpretation would be in line with [21] who argue that tapping–in contrast to other actions such as dragging–requires little motor planning and is often done without thinking about the response. Here, the observed passive advantage does not reflect children's competence, but their performance: Differences in the design of the learning phase affect how children approach the task, which in turn influences their behaviour in the subsequent test phase.

Given the different possible reasons for the findings in Experiment 1, Experiment 2 examined the root of this passive boost. In other words, did active children not learn and correctly recognize the novel word-object associations (relative to the passive children), or did they merely not perform correctly, i.e., not tap on the target object despite knowing what the target object was? We examined this by recording their eye movements as they completed the tapping task in Experiment 2. Here, despite finding a similar passive boost as reported in Experiment 1, we found no evidence for a difference in the time course of active and passive children's recognition of the target object–children in both conditions fixated the target object above chance. While there were some differences found when analyzing eye-movements in each condition separately, the fact that active children fixated the target object at the very least in a similar manner to the passive children suggest that the differences found in the accuracy

measure are unrelated to their competence in word learning but rather with regard to their performance in the tapping task.

Taken together, these results suggest caution in advocating for either a boost in learning when children are allowed to choose what they want to learn [24] or when children are passively presented with new information [16]. At the very least, we found no evidence that there were differences in children's competence across the active and the passive condition. Importantly, we did find differences in children's performance across the two conditions highlighting issues with the design of active learning tasks that may need to be considered in planning digital learning tools. We found that children may have difficulties changing course during the experiment, moving from actively choosing what they want to learn more information about to indicating what they have learned. This was despite the children being told what they needed to do across the different phases of the experiment–and despite there being no such passive boost (at least after the first trials) in familiar trials. In other words, we only found a reliable passive boost in trials where children were tested on their knowledge of the novel word-object associations and not in trials where they were tested on their recognition of highly familiar word-object associations. Thus, it may be that the robust knowledge associated with the familiar objects overrides their prepotent tapping response and the absence of similarly robust knowledge in the novel trials boosts the prepotent tapping response.

We did not find evidence for a difference between children in the active and the passive condition in the 4-AFC task in German children, although we did find such a difference in Malaysian children. We suggest that it is likely that the sudden increase in difficulty as the number of distractors increased from one to three might have had an impact on children, overriding differences across some children in this task. Nevertheless, a passive boost in performance may be expected at some ages in even such a task. Indeed, visual inspection of the data from the German 30-month-olds (similar age to the Malay children) suggests a potential passive boost in all trials but one.

Lured by the bold claims some educational apps make, parents of young children might be tempted to download a large number of apps in the hope of fostering their children's learning in various domains. However, the current study adds to the growing body of evidence that these claims should be taken with caution, since the apps may not be adequately tapping into children's learning progress. Depending on how an educational app is structured, it places the child in the role of an active, self-guided learner. While there is evidence that children can benefit from active learning in some circumstances, the present study paints a different picture. We suggest that an active boost or a passive boost is highly contingent on the task structure and taking this further, the app structure. Depending on the structure of the learning experience, an active choice might actually decrease children's performance in certain tasks, without having much impact on their learned competence. Thus, the attentional and locomotor constraints specific to touchscreen usage should be kept in mind when talking about learning from interactive touchscreen media.

## Supporting information

**S1 Appendix.**
(DOCX)

**S2 Appendix.**
(DOCX)

**S3 Appendix.**
(DOCX)

## Acknowledgments

We thank the families and children who participated in the studies. We would also like to thank Elisabeth Fechner, Meike Förster, Christina Keller, Katharina Reimitz, Juliane Schaarschmidt, Anna Spielvogel, Judith Stolla, and Claudia Wasmuth for helping with data collection.

## Author Contributions

**Conceptualization:** Nivedita Mani, Julien Mayor.

**Formal analysis:** Lena Ackermann, Chang Huan Lo.

**Investigation:** Lena Ackermann.

**Methodology:** Lena Ackermann, Chang Huan Lo.

**Software:** Chang Huan Lo.

**Supervision:** Nivedita Mani, Julien Mayor.

**Visualization:** Lena Ackermann, Chang Huan Lo.

**Writing – original draft:** Lena Ackermann, Chang Huan Lo.

**Writing – review & editing:** Lena Ackermann, Chang Huan Lo, Nivedita Mani, Julien Mayor.

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
