## [Decision Letter · Decision Letter 0]

7 Nov 2019

PONE-D-19-23077

Word learning from a tablet app: 30-month-olds perform better in a passive context

PLOS ONE

Dear Ms. Ackermann,

Thank you for submitting your manuscript to PLOS ONE and apologies for the delay in sending the editorial decision – this was due to the need to invite additional reviewer because the initial reviewers were unable to complete their reviews.

After careful consideration, we feel that it has merit but does not fully meet PLOS ONE’s publication criteria as it currently stands. Therefore, we invite you to submit a revised version of the manuscript that addresses the points raised during the review process.

Two reviewers. who are both experts in the field of children’s technologically assisted learning, agree – and I concur – that your study is timely and addresses a very important topic. However, both reviewers express concerns with respect to how your findings are to be interpreted. I therefore think that in the present form, your study raises more questions than it answers and cannot stand on its own without further investigation.

As indicated by Reviewer 1, there are different aspects to active learning making a stronger theoretical justification of the particular choice of activity (tapping to select items for learning) implemented here necessary.

Furthermore, speculation about what underpins the benefit observed in the passive group needs be constrained through further investigation, as Reviewer 1 points out. Which aspects of your study are germane to active learning and which ones are artefacts of the methodology? In the Discussion, you mention a range of explanations for why tapping may have caused a detrimental effect in the active group (draining resources, having dual function of indicating preference vs. comprehension, prepotent behaviour). Both reviewers also identify other confounds such as potential differences in the pragmatics of the instruction, and differences in the degree of novelty and function of tapping between groups that should be ruled out in a second experiment.

Although I fully appreciate the practical problems associated with testing small children I also share Reviewer 1’s concern about the limited sample size, especially given the amount of fixed effects in your models. In addition to this, further information and possible revision of the statistical models is required (e.g. Did you use a logit link function for the accuracy data? Did you include random slopes [i.e. a maximal random effect structure] and if not why not?)

In short, a major revision should involve one or several further experiments to clarify further what accounts for the inferior performance of the active group. Such a revision should also address all the other concerns expressed by the reviewers.

To enhance the reproducibility of your results, we recommend that if applicable you deposit your laboratory protocols in protocols.io, where a protocol can be assigned its own identifier (DOI) such that it can be cited independently in the future. For instructions see: http://journals.plos.org/plosone/s/submission-guidelines#loc-laboratory-protocols

We look forward to receiving your revised manuscript.

Kind regards,

Vera Kempe

Academic Editor

PLOS ONE

Journal Requirements:

Reviewers' comments:

Reviewer's Responses to Questions

**Comments to the Author**

1. Is the manuscript technically sound, and do the data support the conclusions?

Reviewer #1: Partly

Reviewer #2: Partly

2. Has the statistical analysis been performed appropriately and rigorously? 

Reviewer #1: No

Reviewer #2: Yes

3. Have the authors made all data underlying the findings in their manuscript fully available?

Reviewer #1: Yes

Reviewer #2: No

4. Is the manuscript presented in an intelligible fashion and written in standard English?

Reviewer #1: Yes

Reviewer #2: Yes

5. Review Comments to the Author

Reviewer #1: The authors sought to investigate the effects of interactive features of child-directed tablet apps on 30-month-olds’ word-learning outcomes. Following up on previous work, the authors aimed at evaluating the effects of interactive features while minimizing the confounding effects of stimuli presentation order that could have affected the outcomes of the earlier research. The authors employed a yoked design, in which each pair of age-matched participants was exposed to the same order and duration of stimuli presentation (set by the active condition participants). Consistent with the authors’ predictions, the analyses show a main effect of condition on toddlers’ accuracy on the test trials, suggesting that the interactive features of the app did not facilitate but rather interfered with toddler’s word learning. The authors speculate that this effect could have been caused by the novelty of the tapping action or children’s misinterpretation of the test trials as asking to declare their preference rather that testing their knowledge of words.

My overall impression is that this is an interesting paper that potentially makes a valuable contribution to the literature on children’s ability to learn from mobile and tablet apps. Such knowledge can be of value both to the research community and to parents. However, before this work becomes ready for publication, I recommend that the authors make significant changes to the manuscript addressing the concerns outlined below. I would also prefer to see the authors conduct a follow-up study exploring the possibilities they mention in the Discussion. It is indeed possible that the observed results were driven by the novelty of the tapping action or children’s misinterpretation of the task. I think that a study that tests these conjectures would strengthen the authors’ claim and provide additional information about the factors driving children’s learning from interactive tablet apps.

Specific concerns and other comments:

Line 25-26: Either “by” or “due” for both alternatives would sound better

Line 28: that’s a “yoked” design, correct? Perhaps mention this to orient the reader

Broad comment: I think that a distinction should be made between different types of “active” learning. One way to make an app more interactive is by delegating the choice of an image to be labelled to the player. Another way is by introducing an element of (pseudo) socially contingent exchange (e.g., voice feedback to the child’s actions, or a video of a speaker reacting to such actions). There might be differences in learning outcomes, depending on how the “active” learning component is implemented. I suggest including this point in the GD.

Line 63: I see that the idea is to see whether any sort of contingent feedback would work (given that a manipulation of this sort showed an effect before), but I would still suggest mentioning the difference between social and non-social feedback

Line 80: the SES effect does not seem particularly relevant to the narrative; I suggest either dropping this line or explaining the relevance of this finding (e.g., lower-SES children may have a different pattern of attention distribution, or different access to tablet devices, or something else).

Lines 77-83: No need to cite again, it’s clear that the same study is being referenced throughout the paragraph (same goes to other paragraphs).

General suggestion about the intro: I think the authors are trying to be careful and diligent with citing earlier research but it’s making the intro excessively long. I suggest that many non-critical details about earlier studies can be dropped.

Line 114: It’s better to keep the labeling of conditions consistent. In the abstract the children in the “yoked” condition are referred to as “passive” participants (also, see Line 123). I suggest mentioning that a yoked design was used and then refer to the conditions as “active” and “passive” respectively.

Lines 121-122: What is the prediction for the active group?

Line 140: It would be nice to cite some vocabulary development norms. I am sure that most 30-month-olds know the names of all objects used in the study, but it would be better to rely on some formal evidence.

Line 143: It would be good to see a brief description of such phonotactic constraints. Perhaps, in an Appendix?

Line 164: Did all children stay engaged with the app the entire time? If not, then the differences in the level of engagement between the active and the passive participants should be mentioned (e.g., the passive participants could have found the game less fun and looked away more often).

Line 180: N of participants on the pilot trials?

Line 183: Please include the carrier phrase used

Line 191: Remove the superscript in “was”?

Line 208: How many outliers were there in each condition? Any age-related patterns?

Line 220: It seems to be a lot of variables for a sample of 34 children (the same goes to the analyses of accuracy), I wonder if it would make more sense to run preliminary analyses for variables such as age, gender, and trial order and to only include the variables that do show significant effects to the subsequent analyses. Also, I am not sure why gender but not age is included as a variable. The authors talked about the significance of age for children’s ability to benefit from interactive apps and did not highlight the role of gender. Perhaps, it would make sense to split the sample by age (i.e., “younger half” & “older half”) and test this effect instead of gender.

Line 287: I agree. I wonder if it would make sense to run a follow-up, in which children are given some practice for ~7 days prior to the study so that they do not have to master a new skill along with learning or be too carried away with the thrill of tapping on the screen.

Line 299: It reads as though the children were the ones who did not find the effect of condition. Consider rephrasing (e.g., “Despite . . . , we observed that . . .”).

Reviewer #2: The authors conducted an experimental study to examine the impact of interactivity on toddlers’ (N = 34; 28-36 months) word learning from video. Children were assigned to either actively choose objects that they want to learn labels (active condition) or be exposed to the same labels selected by children in the active condition (passive condition) during the 4 learning trials. Following the 6 familiarization trials where both groups were allowed to interact with the screen, children’s reaction time and accuracy of word learning was measured using 2 and 4 alternative force choice tests (AFC, 12 and 8 trials, respectively). Results showed a significant condition difference in the 2-AFC test phase and a significant condition x order interaction in the 4-AFC test phase. There was no-significant condition effect on reaction time. The authors interpreted the results as evidence to support the adverse effect of interactivity on word learning from video.

Given the prevalence of interactive mobile media in the lives of very young children, it is important to understand the impact of interactivity on learning from screen media. I agree with the authors that there is a need to systematically examine children’s active learning in the context of interactive digital media, particularly in infancy and toddlerhood, as this age groups have shown to be susceptible to video deficit effects. While prior work has examined the impact of either socially adaptive or physically contingent interaction with screens, it remains an open question as to whether the active selection of information benefits toddlers' learning from screens. The authors’ study is addressing this gap in the field by asking children not only to interact with video but also to “choose” what to learn.

Having said that, I think the study needs further justification or clarification regarding potential confounding factors (prompt, familiarization trial). First, in the active condition, the prompt was given as a form of “instruction” (i.e., "Look, here are two pictures. You can tap on one", "Tap on an object, then you’ll hear its name"). On the other hand, in the passive condition, the prompt was given as a form of “question” (i.e., “Do you see the two pictures? Are they beautiful?”). Existing evidence suggests that toddlers appear to utilize social cues such as referential intent of speakers for word learning (Bloom, 2000; Golinkoff & Hirsh-Pasek, 2006). For example, Luchkina, Sobel, & Morgan (2016) showed that 18-month-olds are sensitive to the difference between making statements and asking questions in learning novel labels. Research with preschoolers has shown that explicit instruction might help immediate learning but limit further exploration compared to pedagogical questioning (Bonawitz et al., 2011; Jean et al., 2019). I think further justification is required to conclude that the difference in word learning is due to interactivity rather than the difference in children’s inference of speaker intentions.

Second, both active and passive groups were allowed to interact with the screen during the familiarization phase, which occurred right before the 2 and 4- AFC testing phases. Thus, it is not yet clear whether the active group’s lower performance in testing was due to their active selection during learning or the relatively higher frequencies of the overall physical interactions with the screen. The active group was expected to utilize two functions of tapping: selecting what to learn during learning and providing correct answers during testing. On the other hand, the passive group use tapping mainly to choose their answers during testing. This difference might have contributed to different levels of cognitive loads between the two conditions. In other words, interactivity during the familiarization phase might have benefited the passive group through the goldilocks of interactivity – not too much and not too little but just right – to prepare children to choose correct answers. I think the authors should address this point in the manuscript. Although the authors discussed the findings based on cognitive resources available during encoding and prepotent motor responses in the discussion section, it was not clear to me how they interpreted the interactions available for both conditions during the familiarization phase.

6. PLOS authors have the option to publish the peer review history of their article (what does this mean?). If published, this will include your full peer review and any attached files.

Reviewer #1: No

Reviewer #2: No

---

## [Author Response · Author response to Decision Letter 0]

8 May 2020

As indicated by Reviewer 1, there are different aspects to active learning making a stronger theoretical justification of the particular choice of activity (tapping to select items for learning) implemented here necessary.

Furthermore, speculation about what underpins the benefit observed in the passive group needs be constrained through further investigation, as Reviewer 1 points out. Which aspects of your study are germane to active learning and which ones are artefacts of the methodology? In the Discussion, you mention a range of explanations for why tapping may have caused a detrimental effect in the active group (draining resources, having dual function of indicating preference vs. comprehension, prepotent behaviour). Both reviewers also identify other confounds such as potential differences in the pragmatics of the instruction, and differences in the degree of novelty and function of tapping between groups that should be ruled out in a second experiment.

Although I fully appreciate the practical problems associated with testing small children I also share Reviewer 1’s concern about the limited sample size, especially given the amount of fixed effects in your models. In addition to this, further information and possible revision of the statistical models is required (e.g. Did you use a logit link function for the accuracy data? Did you include random slopes [i.e. a maximal random effect structure] and if not why not?)

- We have now provided more information about the model, clearly specifying the model used. As we note below, we have also now tested several more children (n=132 in Experiment 1, n=38 in Experiment 2), thereby addressing the issue of limited sample size. We have also removed the fixed effects of test.order and gender as requested by Reviewer 1.

In short, a major revision should involve one or several further experiments to clarify further what accounts for the inferior performance of the active group. 

- We have now tested several more children (132 children in Experiment 1 and 38 children in Experiment 2) to examine the developmental time course of the passive benefit we reported in the earlier manuscript (new Experiment 1) as well as to examine a more implicit measure of recognition performance (new Experiment 2). These studies suggest that there is no evidence for differences in the learned competence of children assigned to the active or the passive condition. Both active and passive children recognized the target object successfully and fixated the target object similarly as indexed by the gaze data collected. However, we replicate the performance deficit, with active children showing poorer accuracy of target recognition in the tapping task, across all children tested, regardless of age or cultural background. Taken together, we suggest that these results suggest caution in advocating for an active or a passive boost in digital learning tasks.

Reviewers' comments:

Reviewer #1: 

I would also prefer to see the authors conduct a follow-up study exploring the possibilities they mention in the Discussion. It is indeed possible that the observed results were driven by the novelty of the tapping action or children’s misinterpretation of the task. I think that a study that tests these conjectures would strengthen the authors’ claim and provide additional information about the factors driving children’s learning from interactive tablet apps.

- See response to Editor above. We have now included a second experiment which includes implicit gaze data to examine the extent to which the passive boost found in Experiment 1 was related to a competence deficit in children (active children did not learn the novel word-object associations as well as the passive children) or a performance deficit (active children were carried away by the thrill of tapping on the screen). We found no evidence for a competence deficit and suggest that the results are best explained by a performance deficit. Taken together, we suggest that these results suggest caution in advocating for an active or a passive boost in digital learning tasks.

Specific concerns and other comments:

Line 25-26: Either “by” or “due” for both alternatives would sound better

- Done

Line 28: that’s a “yoked” design, correct? Perhaps mention this to orient the reader

- Done

Broad comment: I think that a distinction should be made between different types of “active” learning. One way to make an app more interactive is by delegating the choice of an image to be labelled to the player. Another way is by introducing an element of (pseudo) socially contingent exchange (e.g., voice feedback to the child’s actions, or a video of a speaker reacting to such actions). There might be differences in learning outcomes, depending on how the “active” learning component is implemented. I suggest including this point in the GD.

Line 63: I see that the idea is to see whether any sort of contingent feedback would work (given that a manipulation of this sort showed an effect before), but I would still suggest mentioning the difference between social and non-social feedback

- We have now added explicit reference to social feedback, pseudo-social contingent feedback and active learning with regards to allowing children to choose the kinds of information they want to learn in the Introduction. 

Line 80: the SES effect does not seem particularly relevant to the narrative; I suggest either dropping this line or explaining the relevance of this finding (e.g., lower-SES children may have a different pattern of attention distribution, or different access to tablet devices, or something else).

- We have now removed this sentence from the manuscript

Lines 77-83: No need to cite again, it’s clear that the same study is being referenced throughout the paragraph (same goes to other paragraphs).

- Done

General suggestion about the intro: I think the authors are trying to be careful and diligent with citing earlier research but it’s making the intro excessively long. I suggest that many non-critical details about earlier studies can be dropped.

- We have now attempted to remove all excessive details about studies while citing earlier research as extensively as possible. 

Line 114: It’s better to keep the labeling of conditions consistent. In the abstract the children in the “yoked” condition are referred to as “passive” participants (also, see Line 123). I suggest mentioning that a yoked design was used and then refer to the conditions as “active” and “passive” respectively.

- We have now changed this to keep consistent. We first mention that it is a yoked design and then continue to only use active and passive respectively. 

Lines 121-122: What is the prediction for the active group?

- We have now outlined the predictions for the active and passive groups across all the age-groups tested.

Line 140: It would be nice to cite some vocabulary development norms. I am sure that most 30-month-olds know the names of all objects used in the study, but it would be better to rely on some formal evidence.

- We have now included data from German vocabulary norms that over 75% of 24-month-olds and close to 100% of 30-month-olds produce the familiar words tested. While German norms do not assess receptive competence, it is therefore likely that the children tested in the current study are familiar with the words presented in the familiar trials. 

Line 143: It would be good to see a brief description of such phonotactic constraints. Perhaps, in an Appendix?

- We have now added an appendix that provides phonetic transcriptions of all pseudowords and how they confirm to the phonotactic rules of German and for Experiment 2, Malay.

Line 164: Did all children stay engaged with the app the entire time? If not, then the differences in the level of engagement between the active and the passive participants should be mentioned (e.g., the passive participants could have found the game less fun and looked away more often).

- We have now added a separate study that allows us to examine this in more detail. In the first study, we did not record looking time data which would allow us to examine the times spent by active and passive children looking at the trials. In the second study, however, we examined the amount of time that children spent looking at the object during the learning phase across conditions. We found that passive participants looked at the target significantly more than active participants at a later window in the learning trial (from 7200ms to 9800ms). To account for this difference, we entered looking time as the target during learning as a predictor of performance at test to account for any potential differences in recognition performance across conditions. This predictor did not explain variability in the recognition performance, suggesting that even though there were differences in the looking times to target (albeit very late in the trial), this looking time difference did not influence recognition performance across conditions. 

Line 180: N of participants on the pilot trials?

- We have now added information of the number of participants in the pilot test. We tested 42 children in this earlier task and realized we need to include a familiar phase to allow passive participants to practice tapping the screen. 

Line 183: Please include the carrier phrase used

- Done. 

Line 191: Remove the superscript in “was”?

- Done

Line 208: How many outliers were there in each condition? Any age-related patterns?

- We now report the outliers split by condition and age-group. 

Line 220: It seems to be a lot of variables for a sample of 34 children (the same goes to the analyses of accuracy), I wonder if it would make more sense to run preliminary analyses for variables such as age, gender, and trial order and to only include the variables that do show significant effects to the subsequent analyses. Also, I am not sure why gender but not age is included as a variable. The authors talked about the significance of age for children’s ability to benefit from interactive apps and did not highlight the role of gender. Perhaps, it would make sense to split the sample by age (i.e., “younger half” & “older half”) and test this effect instead of gender.

- We have now increased the sample size to 132 children with 44 children in each age-group. In addition, we have run the models without the variables test.order and gender report these results. We also include age as suggested by the reviewer to the model and indeed do find an interaction between Condition (active.passive) and age in the 2-AFC task which we break down further. We thank the reviewer for this suggestion.

Line 287: I agree. I wonder if it would make sense to run a follow-up, in which children are given some practice for ~7 days prior to the study so that they do not have to master a new skill along with learning or be too carried away with the thrill of tapping on the screen.

- We have now included a second experiment which includes implicit gaze data to examine the extent to which the passive boost found in Experiment 1 was related to a competence deficit in children (active children did not learn the novel word-object associations as well as the passive children) or a performance deficit (active children were carried away by the thrill of tapping on the screen). We found no evidence for a competence deficit and suggest that the results are best explained by a performance deficit. Taken together, we suggest that these results suggest caution in advocating for an active or a passive boost in digital learning tasks. 

Line 299: It reads as though the children were the ones who did not find the effect of condition. Consider rephrasing (e.g., “Despite . . . , we observed that . . .”).

Based on your recommendation we have now removed trial order from the model and have consequently removed this passage. 

Reviewer #2: 

Having said that, I think the study needs further justification or clarification regarding potential confounding factors (prompt, familiarization trial). First, in the active condition, the prompt was given as a form of “instruction” (i.e., "Look, here are two pictures. You can tap on one", "Tap on an object, then you’ll hear its name"). On the other hand, in the passive condition, the prompt was given as a form of “question” (i.e., “Do you see the two pictures? Are they beautiful?”). Existing evidence suggests that toddlers appear to utilize social cues such as referential intent of speakers for word learning (Bloom, 2000; Golinkoff & Hirsh-Pasek, 2006). For example, Luchkina, Sobel, & Morgan (2016) showed that 18-month-olds are sensitive to the difference between making statements and asking questions in learning novel labels. Research with preschoolers has shown that explicit instruction might help immediate learning but limit further exploration compared to pedagogical questioning (Bonawitz et al., 2011; Jean et al., 2019). I think further justification is required to conclude that the difference in word learning is due to interactivity rather than the difference in children’s inference of speaker intentions.

- We have now included a second experiment which includes implicit gaze data to examine the extent to which the passive boost found in Experiment 1 was related to a competence deficit in children (active children did not learn the novel word-object associations as well as the passive children) or a performance deficit (active children were carried away by the thrill of tapping on the screen). We found no evidence for a competence deficit and suggest that the results are best explained by a performance deficit, i.e., due to the interactivity of the task.

Second, both active and passive groups were allowed to interact with the screen during the familiarization phase, which occurred right before the 2 and 4- AFC testing phases. Thus, it is not yet clear whether the active group’s lower performance in testing was due to their active selection during learning or the relatively higher frequencies of the overall physical interactions with the screen. The active group was expected to utilize two functions of tapping: selecting what to learn during learning and providing correct answers during testing. On the other hand, the passive group use tapping mainly to choose their answers during testing. This difference might have contributed to different levels of cognitive loads between the two conditions. In other words, interactivity during the familiarization phase might have benefited the passive group through the goldilocks of interactivity – not too much and not too little but just right – to prepare children to choose correct answers. I think the authors should address this point in the manuscript. Although the authors discussed the findings based on cognitive resources available during encoding and prepotent motor responses in the discussion section, it was not clear to me how they interpreted the interactions available for both conditions during the familiarization phase.

- See responses to comment above. We suggest that since we do not find difference in children’s target looking in the active and the passive condition, that it is likely that both children learned the word-object associations equally well. Rather, there appears to be a deficit in showing their successful learning that led to the differences in response time and accuracy data

---

## [Decision Letter · Decision Letter 1]

4 Jun 2020

PONE-D-19-23077R1

Word learning from a tablet app: Toddlers perform better in a passive context

PLOS ONE

Dear Dr. Ackermann,

Thank you for submitting the revised version of your manuscript and for making the effort to gather additional data. I was able to solicit reviews from the same reviewers who reviewed the original submission. Both reviewers are pleased to see how you have taken on board their suggestions, and they both provide further recommendations for how to improve the clarity of the paper. In addition, my reading of your revisions leads me to query some aspects of data presentation and to make some suggestions of my own designed to make sure that the data are presented in a way that can withstand careful scrutiny from readers. I trust that the Reviewers’ and my suggestions can be addressed, mainly by revisiting the analyses and figures, and am therefore requesting just a Minor Review.

Experiment 1:

1. You provide the structure of your models which is very helpful but there is no justification for why you decided to omit the random slopes from the models. The random slopes of Age are especially important in the Familiarisation phase were children are presented with real words and their knowledge of these may differ across age groups. If you tried the maximal random effect structure and the models did not converge then please state so explicitly.

2. Unfortunately, it is difficult to appreciate the effects of Condition and Age because your figures do not correspond to the models as they report the data split by Trial Number. As a result, the main effects and interactions are not clear from the figures. As requested by Reviewer 1, please provide figures that show the aggregate effect of Condition and Age for each experimental stage and dependent variable – a format that depicts not just measures of central tendency and variability but also the shape of the distribution such as violin or raincloud plots would be most informative but this is merely a suggestion. When addressing the dynamics of performance over time in your text (e.g. by referring to early vs. later trials) the current figures can be referred to in an Appendix or Supplementary Materials.

3. My last point is potentially serious unless I misunderstood your reporting (in which case I apologise): I wasn’t sure whether you are reporting the correct reference category. For example, in the RT model, if in the Familiarity phase children in the Active condition were faster as you suggest in line 228, and Passive was the reference condition, I would expect the slope estimate (beta) to be negative. Similarly, for 2AFC RTs, if passive children were faster and Passive is the reference category then beta should be positive if the active children were slower. A similar discrepancy arises in the Accuracy analyses. Please check whether the reference category corresponds to the reported direction of effects. Also, please clarify whether you are using sum or treatment coding as this has implications for interpreting the intercept. My big concern is that if Passive was indeed the reference category then your models show exactly the opposite of what you claim.

Experiment 2:

1. Similarly, as requested for Experiment 1, please provide a graph for the learning phase with overall fixation proportions that illustrate the main effect mentioned in lines 443/444, i.e. something similar to Figure 7.

2. For the mixed-effect models, you state that you used sum-coding. Was sum coding also used for the RT and accuracy analyses in Experiment 1? Please clarify. Again, there is confusion with the estimates: if Passive is the reference category then a positive beta for Condition (e.g. 0.219 for 2AFC) would suggest accuracy is higher in the active group? This is not what Figure 10 suggests so is it possible that Active was the reference category? Again, this may have serious implications for your results so I hope that your reporting of the reference category was just a simple error and Active is the reference category (this is likely because lme4 takes the category that comes first in the alphabet as reference category unless specified otherwise). Please check and clarify.

We look forward to receiving your revised manuscript.

Kind regards,

Vera Kempe

Academic Editor

PLOS ONE

Reviewers' comments:

Reviewer's Responses to Questions

**Comments to the Author**

1. If the authors have adequately addressed your comments raised in a previous round of review and you feel that this manuscript is now acceptable for publication, you may indicate that here to bypass the “Comments to the Author” section, enter your conflict of interest statement in the “Confidential to Editor” section, and submit your "Accept" recommendation.

Reviewer #1: All comments have been addressed

Reviewer #2: All comments have been addressed

2. Is the manuscript technically sound, and do the data support the conclusions?

Reviewer #1: Partly

Reviewer #2: Yes

3. Has the statistical analysis been performed appropriately and rigorously? 

Reviewer #1: Yes

Reviewer #2: Yes

4. Have the authors made all data underlying the findings in their manuscript fully available?

Reviewer #1: Yes

Reviewer #2: Yes

5. Is the manuscript presented in an intelligible fashion and written in standard English?

Reviewer #1: Yes

Reviewer #2: Yes

6. Review Comments to the Author

Reviewer #1: This is a substantially improved manuscript. I am pleased to see the reorganization of the intro, the much clearer description of the analyses, and the details of the procedure omitted in the previous submission. The authors added a substantial number of participants to Experiment 1, which increased the statistical power of their analyses, and conducted Experiment 2 to help disambiguate between the potential explanations of children’s behavior in Experiment 1. Experiment 2 also offers an advantage of generalizing the effect found in Experiment 1 to a different culture.

What remains unclear to me is :

(1) how the authors decided on how many participants to add to Experiment 1

(2) why Experiment 2 was specifically conducted with Malay-speaking children

(3) how might the reported experiments contribute to the discussion on the effects of children’s choice of what they learn on learning outcomes (cf Sim et al.) and what the developmental trajectory of these effects might look like

(4) why children’s frequency and duration of lookaways in Experiment 2 wasn’t analyzed: this would have provided us with some insight about the potential differences in children’s engagement with the task and their effects on word learning (if any).

I suggest that the authors address these questions in the next round of revision.

Other points:

Line 24: please clarify what you mean by “passive in such situations”.

Line 54: The transition to talking about the video deficit is abrupt. Screen media does not necessarily imply *passive viewing* in the context of tablets. Consider adding an opening sentence that will orient the reader and help expect the discussion about the video deficit.

Line 64: Define pseudo-social. I can infer from my previous comments what this means, but the reader will need an explicit definition.

Lines 71-73: Specify the ages of “younger toddlers” and “older toddlers”.

Table 1: Consider renaming 36-mo group a “40-mo group”. The average age is 39.62 months.

Line 163: Perhaps child-directed speech? Infant-directed speech seems strange here.

Line 193: Remove superscript from “was”.

Table 2: Display mean reaction times separately by condition.

Lines 269-271: I am having trouble parsing this sentence. Please consider rephrasing. Something like “In this experiment we set out to examine whether the opportunity to choose the objects that will be labeled influences children’s learning of those labels”.

Line 288: So, this effect seems to be pretty small and relatively hard to interpret? Perhaps discuss if anything can be made of this? There was a discussion in the intro about the effects of age, which, I believe, should continue in the results section of Experiment 1. Does the observed condition*age interaction inform our understanding of the developmental trajectory of children’s ability to benefit from choosing what they learn?

Line 296: Remove the hyphen from age groups.

Line 385: Remove “both”.

Lines 406-431: It seems to me that considering that the level of engagement with the app could be a potential factor that influences children’s learning, it would be reasonable to analyze the frequency and the duration of lookaways in each condition. Looking to the target object during the learning phase does tap into this question but it misses out on children’s overall level of engagement with the task.

Lines 619-630: Unclear to me if this is quite the right conclusion. If children learn regardless of the condition, it suggests that the level of app interactivity (at least as far as controlling the information presented) is not a predictor of learning outcomes. The way I see it, this is not a good reason for parents or educators to be alarmed.

Reviewer #2: Thank you for your careful attention to my comments and questions. The authors provided further details to clarify the points in the previous reviews, including my own. I also appreciate the authors' effort to increase the sample size and conduct a follow-up study to address potential confounders. I am satisfied with how the authors have addressed and incorporated my comments. I have one suggestion describe below, to improve the clarity of the manuscript.

Page 13: In the discussion section of Experiment 1, the authors said, "there were no differences in children’s accuracy of responses" (Page 13, L277-278) during the familiar test phase. However, the results section of Experiment 1 includes the following: "the familiar phase and the 2-AFC, condition significantly predicts accuracy, with children providing more accurate responses in the passive condition relative to the active condition" (Page 13, L260-261). These two statements contradict each other, so it should be corrected.

7. PLOS authors have the option to publish the peer review history of their article (what does this mean?). If published, this will include your full peer review and any attached files.

Reviewer #1: No

Reviewer #2: No

---

## [Author Response · Author response to Decision Letter 1]

17 Jul 2020

Experiment 1:

1. You provide the structure of your models which is very helpful but there is no justification for why you decided to omit the random slopes from the models. The random slopes of Age are especially important in the Familiarisation phase were children are presented with real words and their knowledge of these may differ across age groups. If you tried the maximal random effect structure and the models did not converge then please state so explicitly.

Thank you for pointing this out. The earlier model did not include age as a random effect as the initial dataset only included data from one age. However, we updated the model accordingly and ran into convergence issues when we tried to fit a maximal random effects structure, which we have now also reported in the paper.

2. Unfortunately, it is difficult to appreciate the effects of Condition and Age because your figures do not correspond to the models as they report the data split by Trial Number. As a result, the main effects and interactions are not clear from the figures. As requested by Reviewer 1, please provide figures that show the aggregate effect of Condition and Age for each experimental stage and dependent variable – a format that depicts not just measures of central tendency and variability but also the shape of the distribution such as violin or raincloud plots would be most informative but this is merely a suggestion. When addressing the dynamics of performance over time in your text (e.g. by referring to early vs. later trials) the current figures can be referred to in an Appendix or Supplementary Materials.

Thank you for the suggestion to include figures that depict the distribution of data. We have now replaced Fig. 3 with a violin plot that shows children’s reaction times in each test phase, split by age and condition. Similarly, Fig. 4 has been replaced by a plot showing the mean accuracy for each age group and test phase. The plots by trial have been moved to an appendix.

3. My last point is potentially serious unless I misunderstood your reporting (in which case I apologise): I wasn’t sure whether you are reporting the correct reference category. For example, in the RT model, if in the Familiarity phase children in the Active condition were faster as you suggest in line 228, and Passive was the reference condition, I would expect the slope estimate (beta) to be negative. Similarly, for 2AFC RTs, if passive children were faster and Passive is the reference category then beta should be positive if the active children were slower. A similar discrepancy arises in the Accuracy analyses. Please check whether the reference category corresponds to the reported direction of effects. Also, please clarify whether you are using sum or treatmen t coding as this has implications for interpreting the intercept. My big concern is that if Passive was indeed the reference category then your models show exactly the opposite of what you claim.

Thank you for pointing this out. We have now clarified what coding scheme was used and how this relates to the intercept and the beta estimates. In our sum coding scheme, passive is coded as 1 and active as -1. A positive beta, as in the model for the familiar phase, thus indicates that the condition coded as 1 (passive) has a higher RT than the intercept (mean RT ignoring effect of condition). We opted for sum coding instead of treatment coding as this makes the intercept interpretable as an overall measure of performance regardless of condition.

Experiment 2:

1. Similarly, as requested for Experiment 1, please provide a graph for the learning phase with overall fixation proportions that illustrate the main effect mentioned in lines 443/444, i.e. something similar to Figure 7.

We have now provided a graph for the learning phase.

2. For the mixed-effect models, you state that you used sum-coding. Was sum coding also used for the RT and accuracy analyses in Experiment 1? Please clarify. Again, there is confusion with the estimates: if Passive is the reference category then a positive beta for Condition (e.g. 0.219 for 2AFC) would suggest accuracy is higher in the active group? This is not what Figure 10 suggests so is it possible that Active was the reference category? Again, this may have serious implications for your results so I hope that your reporting of the reference category was just a simple error and Active is the reference category (this is likely because lme4 takes the category that comes first in the alphabet as reference category unless specified otherwise). Please check and clarify.

Thank you for pointing this out. See our response above. We have now clarified this in the paper.

6. Review Comments to the Author

Reviewer #1: 

What remains unclear to me is :

(1) how the authors decided on how many participants to add to Experiment 1

Thank you for this question. We had initially decided to test 20 children in each condition in each age-group (i.e., 40 children in each age-group). However, due to an error, we ended up testing 22 children in one condition in one age-group. We, therefore, increased samples across age-groups to test 22 children in each age-group. Posthoc power analyses revealed that this was adequate for our main tests. 

(2) why Experiment 2 was specifically conducted with Malay-speaking children

This was a convenience sample given that two of the authors were located in Malaysia at the time of data collection. Indeed, this was the same reason that we tested German children in the initially submitted study. We do not place any value on the origin of the children tested in the study, in keeping with the criticism of the WEIRD focus in the literature, which does not typically require similar justification of why predominantly Western Educated children are tested in most studies in the field. We merely wish to highlight with a different sample of children that we replicate the results in children from two different cultural and language backgrounds and hope by this to contribute to the diversity of the findings reported. We have now made this clear in the article. 

(3) how might the reported experiments contribute to the discussion on the effects of children’s choice of what they learn on learning outcomes (cf Sim et al.) and what the developmental trajectory of these effects might look like

We thank the reviewer for pointing us to this paper. This is a very interesting study, which, nevertheless is at a tangent to the current paper. In this study, the authors create, to a certain extent, an information gap which can be resolved by children in their active selection of objects that could close this information gap. Similar work with children in a word learning study (Zettersten & Saffran, 2019) finds that children actively choose objects that reduce the information gap, similar to Sim et al. (2015) who find in a non-word learning paradigm that children sample objects that help them determine the categorisation of objects to different houses. In our study, however, we did not create a similarly imbalanced information gap, at the very least, there was no informational gain, apart from salience or children’s preference for one object over the other, which would bias children to choose one object over the other. Therefore, it is difficult to consider the development of such effects using such disparate paradigms. We have however noted the two studies and their findings and the relation to the current work in the Introduction. 

(4) why children’s frequency and duration of lookaways in Experiment 2 wasn’t analyzed: this would have provided us with some insight about the potential differences in children’s engagement with the task and their effects on word learning (if any).

We report as is standard in the literature the proportion of target looking to the images during both the training and the test phases. This is by far the most oft-reported index of children’s performance in the literature and through the plotting of both time course graphs in addition to the overall statistics provides an overview of children’s looking behaviour across the trial. In addition, we note that we also plotted the looking time during the learning phase in Figure 6 of the old manuscript (current Figure X). This captures duration of lookaways during the training phase (since the proportion of target looks would drop during at any given time if the child were looking away) and showed a difference between the active and the passive children (in the timewindow between 7200 to 9800ms). We also note that the proportion of target looking in the learning phase was not a significant predictor of participants’ responding in the trial. Given this analyses and the established measure of proportion of target looking that we do report, we did not add a further variable(s) to the analysis. 

I suggest that the authors address these questions in the next round of revision.

Other points:

Line 24: please clarify what you mean by “passive in such situations”.

Done.

Line 54: The transition to talking about the video deficit is abrupt. Screen media does not necessarily imply *passive viewing* in the context of tablets. Consider adding an opening sentence that will orient the reader and help expect the discussion about the video deficit.

Done. We have now made clear that in this context, we refer to a video deficit when children were exposed to training stimuli on screen passively (and explained what this may mean). 

Line 64: Define pseudo-social. I can infer from my previous comments what this means, but the reader will need an explicit definition.

Done. 

Lines 71-73: Specify the ages of “younger toddlers” and “older toddlers”.

Done. 

Table 1: Consider renaming 36-mo group a “40-mo group”. The average age is 39.62 months.

We agree and have changed this throughout the manuscript

Line 163: Perhaps child-directed speech? Infant-directed speech seems strange here.

Done

Line 193: Remove superscript from “was”.

Done 

Table 2: Display mean reaction times separately by condition.

Done

Lines 269-271: I am having trouble parsing this sentence. Please consider rephrasing. Something like “In this experiment we set out to examine whether the opportunity to choose the objects that will be labeled influences children’s learning of those labels”.

Done. We have now changed it as follows: In this experiment, we set out to examine whether being given the opportunity to choose the objects that will be labeled influences children’s learning of these word-object associations in a touchscreen-based word learning task.

Line 288: So, this effect seems to be pretty small and relatively hard to interpret? Perhaps discuss if anything can be made of this? There was a discussion in the intro about the effects of age, which, I believe, should continue in the results section of Experiment 1. Does the observed condition*age interaction inform our understanding of the developmental trajectory of children’s ability to benefit from choosing what they learn?

First, we now make clearer that differences in the familiar test phase are to be treated with caution for the reasons mentioned earlier in the same paragraph. Second, we now make clear that collapsing across the measures we tested, there are no developmental differences in the passive benefit that we find at all ages. Thus, this was a response time boost at the younger age-group and an accuracy boost at the older age group. But in both cases, this was a passive benefit and not as hypothesised an early active benefit changing to a passive benefit. 

Line 296: Remove the hyphen from age groups.

We have done this throughout the manuscript.

Line 385: Remove “both”.

Done

Lines 406-431: It seems to me that considering that the level of engagement with the app could be a potential factor that influences children’s learning, it would be reasonable to analyze the frequency and the duration of lookaways in each condition. Looking to the target object during the learning phase does tap into this question but it misses out on children’s overall level of engagement with the task.

As we note above, the proportional of target looking throughout the learning phase was coded and entered into the model during target recognition and was not a significant predictor of children’s target recognition success. This provides us with children’s engagement during the training phase and suggests that there was no evidence that engagement during the training phase predicts learning. Furthermore, we plot the proportion of target looking during the test trials and the learning trials. This time course captures the duration of lookaways since the proportion of target looking will systematically drop were children to look away at any given point in the trial. We did not examine frequency and duration of lookaways given the established measure that we incorporate into the analyses, namely, the proportion of target looking measure. 

Lines 619-630: Unclear to me if this is quite the right conclusion. If children learn regardless of the condition, it suggests that the level of app interactivity (at least as far as controlling the information presented) is not a predictor of learning outcomes. The way I see it, this is not a good reason for parents or educators to be alarmed.

We have now rephrased this to highlight that the apps may not be accurately tapping into children’s learning progress (as was seen in the case of the active children) depending on how it is structured.

Reviewer #2: 

Page 13: In the discussion section of Experiment 1, the authors said, "there were no differences in children’s accuracy of responses" (Page 13, L277-278) during the familiar test phase. However, the results section of Experiment 1 includes the following: "the familiar phase and the 2-AFC, condition significantly predicts accuracy, with children providing more accurate responses in the passive condition relative to the active condition" (Page 13, L260-261). These two statements contradict each other, so it should be corrected.

 Thank you for pointing this out. We have now corrected this.

---

## [Editor Report · Decision Letter 2]

23 Jul 2020

PONE-D-19-23077R2

Word learning from a tablet app: Toddlers perform better in a passive context

PLOS ONE

Dear Dr. Ackermann,

Thank you for submitting the revised version. Having checked it I do not see any need to send it out for further review as I am basically happy with the revisions. However, I will return it to you requesting Minor Corrections hoping you will be able to reupload it with some small changes as detailed below – all suggested to improve clarity of the analyses and their interpretation. Once you have addressed these points I trust the manuscript should be ready for publication. My line numbers below refer to your version without the tracked changes.

The wording from line 233 onwards (and the equivalent for Exp 2) should be rephrased to explain why there was no random slope of Age by Words. You presently refer to a random effect of Age which is not quite correct as you have random effects of Participant and Word. In essence, you should explain why your model did not contain the full random effect structure

RT_log_ ~ Condition * Age + (1|Participant) + (Condition * Age|Word)

I am not suggesting you change the model but am asking you to state clearly that full random effect structure with slopes of Condition, Age and their interaction by Word resulted in convergence problems. This is just a request to put the wording right.

I also note that in figures A1 and A2 the group now labelled as 40-months-old is still labelled as 36-months-old – please correct this. Also, you currently have two figures 2 – I presume the latter figure labelled 2 on the top left of the page is meant to be figure A2. While you are at this it would help if you could put the figures in the right order to expedite the final editing process.

Line 240: Please change ‘The model included the interaction…’ to ‘The models included the interaction…’ as you are describing three models, one for each task.

I am still somewhat confused about the coding of the effect of Age in the analyses of Experiment 1. The sum-coding of Age adopted now makes it difficult to interpret the coefficients and to ascertain whether performance in the 30-months-olds differed from the 24-months-olds, and whether performance of the 40-months-olds differed from the 30-months-olds. My preference would be for treatment coding with either 24- or 30-months-olds as the reference category but to improve general interpretation of the model results please try to be a bit more explicit in the text regarding the effects of Age in the Familiarisation and 2-AFC tasks.

In your narrative that explains the RT analyses on pages 11/12 please include a brief summary of the model results for the 4-AFC task – Table 3 shows there were no significant effects but that should be stated explicitly in the text.

Please tone down the interpretation in lines 303-305 – while the 24-months-olds were indeed the only ones showing an effect of Condition it is misplaced to call it “much faster” based on what the violin plot in Figure 3 shows. “Somewhat faster” seems more appropriate.

Please label the dashed horizontal line in Figures 6 and 7.

Lines 310 and 314 referring to the ‘passive benefit across the ages’ sound redundant – please rephrase.

To address the points raised by Reviewer 1, please include a brief justification for analysing only looks to the target and not lookaways. You have justified it in your rebuttal letter but other readers may have similar questions so it would be good to address this head-on, somewhere in the vicinity of lines 406-409.

A rebuttal letter that responds to each point raised by the academic editor and reviewer(s). You should upload this letter as a separate file labeled 'Response to Reviewers'. This can be fairly short simply listing my original points and whether you were able to address them.A marked-up copy of your manuscript that highlights changes made to the original version. You should upload this as a separate file labeled 'Revised Manuscript with Track Changes'.An unmarked version of your revised paper without tracked changes. You should upload this as a separate file labeled 'Manuscript'.

I very much look forward to receiving your revised version, which I hope will be the final one.

Kind regards,

Vera Kempe

Academic Editor

PLOS ONE

---

## [Author Response · Author response to Decision Letter 2]

25 Sep 2020

Comment 1: The wording from line 233 onwards (and the equivalent for Exp 2) should be rephrased to explain why there was no random slope of Age by Words. You presently refer to a random effect of Age which is not quite correct as you have random effects of Participant and Word. In essence, you should explain why your model did not contain the full random effect structure

RTlog ~ Condition * Age + (1|Participant) + (Condition * Age|Word)

I am not suggesting you change the model but am asking you to state clearly that full random effect structure with slopes of Condition, Age and their interaction by Word resulted in convergence problems. This is just a request to put the wording right.

Response 1: Note that age was entered into the model as a fixed effect (as mentioned in line 234), not as a random effect. We examine performance here on a 2-AFC and a 4-task here where the object is the target in one trial and the distractor is the other. Therefore, performance on a given word cannot be entirely attributed to the word alone but also the distractor displayed in test trials. Thus, including variation across individual words in the effect of condition*age is confounded by influence of the distractor in each trial. We tried initially to reduce this confound by not running the maximal model. We have now, however, corrected this the following way. We now run models with random effects on the object rather than the word and focus on the leaner models in the manuscript while reporting the results of the full maximal models in Appendix C. We also note that this was our reason for not reporting the maximal models in the paper on page XX. There were a few minor deviations across the lean models and the maximal models which we highlight across the manuscript. In particular, the accuracy results in the 2-AFC and 4-AFC task remain the same across all studies, while differences in the reaction time measure at 24-months disappear. 

Note that the model we run differs from the model suggested by you since we examine the data by pairs (i.e., the active-passive pair) and could therefore include variation in the effect of condition across the random effect of participant pairs. Running the models as suggested by you with participants rather than pairs yields very similar results, but given the design of the study we retain the paired analyses. Finally, in rerunning the analyses we realised that 1 pair in the German study at 24-months and 3 pairs in the Malaysian study were not age-matched and therefore have excluded these children from the new analyses. We apologise for not noticing this sooner. 

Comment 2: I also note that in figures A1 and A2 the group now labelled as 40-months-old is still labelled as 36-months-old – please correct this. Also, you currently have two figures 2 – I presume the latter figure labelled 2 on the top left of the page is meant to be figure A2. While you are at this it would help if you could put the figures in the right order to expedite the final editing process.

Response 2: Many thanks for pointing this out. We have now relabeled both figures and re-ordered the figures. 

Comment 3: Line 240: Please change ‘The model included the interaction…’ to ‘The models included the interaction…’ as you are describing three models, one for each task.

Response 3: Many thanks for pointing this out. We have now updated the sentence accordingly. 

Comment 4: I am still somewhat confused about the coding of the effect of Age in the analyses of Experiment 1. The sum-coding of Age adopted now makes it difficult to interpret the coefficients and to ascertain whether performance in the 30-months-olds differed from the 24-months-olds, and whether performance of the 40-months-olds differed from the 30-months-olds. My preference would be for treatment coding with either 24- or 30-months-olds as the reference category but to improve general interpretation of the model results please try to be a bit more explicit in the text regarding the effects of Age in the Familiarisation and 2-AFC tasks.

Response 4: Thank you for pointing this out. We have now explicitly clarified the interpretation of the models as follows:

RT (Table 3): “In other words, there was a difference in the effect of condition between 24- and 30-months and between 24- and 40-months in the 2-AFC model, but not in the familiar or the 4-AFC model.”

Accuracy (Table 4): “ There was a difference in the effect of condition between 24- and 40-months in the familiar model and between 24- and 30-months in the 2-AFC model. There was no effect of condition on accuracy in the 4-AFC task. There were no interactions between condition and age between any of the other ages tested in any of the other models.”

Comment 5: In your narrative that explains the RT analyses on pages 11/12 please include a brief summary of the model results for the 4-AFC task – Table 3 shows there were no significant effects but that should be stated explicitly in the text.

Response 5. Thank you for pointing this out. We have now explicitly stated this in the text as follows: “Furthermore, there was no effect of condition in the 4-AFC task, nor were there any interactions between condition and age in this task (see Table 3).”

Comment 6: Please tone down the interpretation in lines 303-305 – while the 24-months-olds were indeed the only ones showing an effect of Condition it is misplaced to call it “much faster” based on what the violin plot in Figure 3 shows. “Somewhat faster” seems more appropriate.

Response 6: Many thanks for pointing this out. We have now updated the sentence accordingly. 

Comment 7: Please label the dashed horizontal line in Figures 6 and 7.

Response 7: Both figures are now updated. 

Comment 8: Lines 310 and 314 referring to the ‘passive benefit across the ages’ sound redundant – please rephrase.

Response 8: Many thanks for pointing this out. We have now removed this sentence.

Comment 9: To address the points raised by Reviewer 1, please include a brief justification for analysing only looks to the target and not lookaways. You have justified it in your rebuttal letter but other readers may have similar questions so it would be good to address this head-on, somewhere in the vicinity of lines 406-409.

Comment 9: We have now included the following sentence where suggested by you. “We report, as is standard in the literature, the proportion of target looking to the images during both the training and the test phases. In addition, we plot the looking time during the learning phase and the test phase in Figures 6, 7 and 8. Together these capture not just the proportion of looks to the target but also the look-aways to the distractor since the proportion of target looks would correspondingly drop at any given time were the child to be looking at the distractor than the target.”

---

## [Editor Report · Decision Letter 3]

29 Sep 2020

Word learning from a tablet app: Toddlers perform better in a passive context

PONE-D-19-23077R3

Dear Dr. Ackermann,

We’re pleased to inform you that your manuscript has been judged scientifically suitable for publication and will be formally accepted for publication once it meets all outstanding technical requirements. Thank you very much for your thorough revision and your efforts in trying to address all concerns. It is good to see this important work now being shared widely at PlosOne.

Kind regards,

Vera Kempe

Academic Editor

PLOS ONE
---

## [Editor Report · Acceptance letter]

19 Nov 2020

PONE-D-19-23077R3 

Word learning from a tablet app: Toddlers perform better in a passive context 

Dear Dr. Ackermann:

I'm pleased to inform you that your manuscript has been deemed suitable for publication in PLOS ONE. Congratulations! Your manuscript is now with our production department. 

Kind regards, 

on behalf of

Prof Vera Kempe 

Academic Editor

PLOS ONE